# Adversarial Generalization of Unfolding (Model-based) Networks

**Vicky Kouni**
LAMSADE, Paris Dauphine - PSL Research University
`vasiliki.kouni@dauphine.psl.eu`

## Abstract

Unfolding networks are interpretable networks emerging from iterative algorithms, incorporate prior knowledge of data structure, and are designed to solve inverse problems like compressed sensing, which deals with recovering data from noisy, missing observations. Compressed sensing finds applications in critical domains, from medical imaging to cryptography, where adversarial robustness is crucial to prevent catastrophic failures. However, a solid theoretical understanding of the performance of unfolding networks in the presence of adversarial attacks is still in its infancy. In this paper, we study the adversarial generalization of unfolding networks when perturbed with $l_2$-norm constrained attacks, generated by the fast gradient sign method. Particularly, we choose a family of state-of-the-art overaparameterized unfolding networks and deploy a new framework to estimate their adversarial Rademacher complexity. Given this estimate, we provide adversarial generalization error bounds for the networks under study, which are tight with respect to the attack level. To our knowledge, this is the first theoretical analysis on the adversarial generalization of unfolding networks. We further present a series of experiments on real-world data, with results corroborating our derived theory, consistently for all data. Finally, we observe that the family's overparameterization can be exploited to promote adversarial robustness, shedding light on how to efficiently robustify neural networks.

## 1 Introduction

The advent of deep unfolding networks (DUNs) [17] ushered in a new paradigm for inverse problems [37], by transforming iterative optimization algorithms into trainable neural architectures [39]. The starting point of DUNs is that many of these algorithms can be written compactly in a neural network formulation:

$$x_{k+1} = \text{nonlinearity}(\text{linear\_transform}(x_k) + \text{bias\_term}). \qquad (*)$$

In this paper, we are interested in the inverse problem of Compressed Sensing (CS) [44], modeling a plethora of modern applications, from medical imaging and speech processing, to communication systems and cryptography, where robustness is crucial for ensuring safe and reliable inference.

CS deals with recovering data from missing, *noisy observations*, given that the structure of the data is sparse via some *fixed* transform (e.g. wavelets for images). To address CS, a popular approach relies on formulating it as a LASSO optimization problem, and then using some iterative proximal algorithm to solve it [8]. Since the output of the proximal algorithm at a given iteration can be written as in ($*$), the algorithm's iterations can be treated as the layers of a DNN, so that the algorithm is "unfolded" into said DNN [17]. While all parameters are fixed in the proximal algorithm, its unfolded counterpart is treated as a structured DNN, with *unknown and thus learnable* parameters, e.g., the sparsifying transform. As such, the sparse data model is inherited by DUNs [4, 24, 42], rendering them as interpretable DNNs with superior reconstruction quality in reduced time [40, 41].

39th Conference on Neural Information Processing Systems (NeurIPS 2025).

Interestingly, learnable sparsifying transforms seem to improve the robustness of DUNs against additive noise [25], with recent empirical studies also focusing on DUNs' robustness against gradient-based adversarial attacks [13, 46]. Despite these advances, a solid mathematical understanding of the performance of DUNs in the presence of adversarial attacks, both during training and test times, remains elusive. Generally, adversarial robustness of standard DNNs is a rigorous research topic [18, 31], with adversarial generalization [29] being at the forefront of recent research interest, as it can be leveraged to control and analyze the robustness of DNNs against adversarial attacks like the fast gradient sign method (FGSM) [16], the projected gradient method (PGD) [27], and their variants [22, 26]. A key tool in this direction is the adversarial Rademacher complexity (ARC) [1], which quantifies model complexity under worst-case perturbations, and enables a structured way of studying the adversarial generalization of DNNs.

**Our contributions in the context of related work.** Motivated by recent advances in DUNs and adversarial generalization, in this paper, we seek to address the following core question:

*What is the generalization performance of DUNs in the presence of adversarial attacks?* ($\star$)

To that end, our main contributions are as follows:

- From a broader family of DUNs, we select a state-of-the-art representative, parameterized by an overcomplete sparsifier, leading to an *overparameterized regime*. Then, we perturb that DUN with FGSM attacks – considered both during training and test times – constrained in the $l_2$-norm. *We differentiate our approach from related work in the following aspect: we prove the Lipschitz continuity of the attacked DUN with respect to the learnable overcomplete sparsifier, and deploy this result as a new cornerstone to estimate the ARC of the DUN.*

- We leverage the ARC estimate to upper-bound the adversarial generalization error of the examined DUN; to our knowledge, these are the *first theoretical results on the generalization performance of adversarially perturbed DUNs*. Specifically, our main Theorem, stated informally below, provides a convincing partial answer to ($\star$), and highlights how the overcompleteness of the learnable sparsifier, the number of layers, and the level of the adversarial FGSM-based attack, ripple out to DUN adversarial generalization. To the best of our knowledge, our result improves on state-of-the-art related work [12, 49] in terms of the number of layers and the attack level (cf. Table 1).

  **Theorem** (cf. Theorem 3). *With high probability, the adversarial generalization error of the examined DUN roughly scales like $\sqrt{NL\log(\varepsilon)}$, with L being the total number of layers, N controlling the overcompleteness of the learnable sparsifier, and $\varepsilon$ being the attack level.*

- We evaluate our mathematical results with numerical experiments on real-world data. *Our findings are three-fold*: a) the empirical adversarial generalization of the DUN conforms with the theoretical one, consistently for all data, b) overcompleteness promotes adversarial robustness: the higher the $N$, the more adversarially robust the DUN is to increasing attack levels, c) comparisons with a baseline DUN learning an orthogonal sparsifier highlight the superiority of overcompleteness over orthogonality in the context of adversarial robustness, with our adversarially perturbed DUN outperforming the baseline in all scenarios.

Overall, our proposed study provides the first theoretical framework for understanding and improving the adversarial robustness of DUNs in CS, with direct implications for safety-critical applications. In medical imaging, for example, our results could help ensure that MRI reconstructions remain reliable even under adversarial perturbations, reducing the risk of misdiagnosis. Furthermore, by showcasing that overparameterization via learnable overcomplete sparsifiers improves robustness, our work offers concrete design principles for developing resilient, interpretable networks in real-world deployments, where accuracy, efficiency, and security must coexist.

## 2    Related work

To place our contributions in context, we review relevant literature in the next paragraphs.
**(Adversarial) Robustness in DUNs.** DUNs solve the CS problem in the presence of noise added during the observation process; the same holds true when perturbing DUNs with adversarial attacks. Thus, studying the (adversarial) robustness of DUNs is reasonable. As a starting point, early work [25] highlights the robustness to Gaussian noise added to the observations, for a DUN learning an

**Table 1:** Comparison of our adversarial generalization error bounds to related work's. Bold letters indicate the method yielding the tighter – which is desirable – upper-bounds, in terms of attack level $\varepsilon$ and number of layers $L$. For [12, 49], $\beta_k$ denotes the upper bound on the spectral norm of the parameter matrix at the $k$th layer output. For our work, the upper bound on the spectral norm of the parameter matrix shared across all layers is $\sqrt{\beta}$.

| Method | Surrogate FGSM-based loss [12] | Covering numbers [49] | **Ours (Lipschitz continuity + surrogate FGSM-based loss + covering numbers)** |
|---|---|---|---|
| Adv. gen. error bounds | $O(\varepsilon \prod_{k=1}^{L} \beta_k)$ | $O(\varepsilon \prod_{k=1}^{L} \beta_k)$ | $O(\sqrt{L \log(\beta \varepsilon)})$ |

*overcomplete sparsifier*, i.e., a sparsifying transform with more rows than columns, as opposed to a DUN learning an *orthogonal sparsifier*. This behavior is consistent with the model-based regime, where overcomplete sparsifiers promote robust reconstruction of inverse problems [7]. More on the adversarial robustness of DUNs, [46] develops a new, DUN-specific, adversarial attack, which is compared to FGSM attacks, to elaborate empirically on the robustness of the examined DUN. Additionally, [13] explores experimentally the DUN robustness in the presence of a PGD attack [27]. Despite these advances, a theoretical investigation on the adversarial robustness and generalization of DUNs is still in its infancy, motivating the need for a rigorous framework that can explain and quantify the behavior of DUNs in the presence of adversarial attacks.

**Adversarial learning and generalization.** To develop such a framework for DUNs, we turn to the well-established field of adversarial learning, which aims to improve the robustness of neural networks, by solving a maximization problem that identifies worst-case attacks. A key focus in this area is adversarial generalization, which pertains to the ability of adversarially trained DNNs to generalize well to test-time adversarially perturbed data, and is quantified via the adversarial generalization error. This quantity incorporates a maximization over input perturbations, reflecting the nature of adversarial training. To study the adversarial generalization error, various theoretical tools have been proposed, including on-average stability [48], minimax theory [45], compression arguments [2], PAC-Bayesian theory [43], and adversarial Rademacher complexity (ARC) [1]. The latter is a prominent one, due to its elegance and long-standing connection with machine learning [3]. The ARC is particularly suitable for our study, as it aligns well with the structured architecture of DUNs and offers meaningful complexity measures connected to generalization performance.

**Adversarial Rademacher complexity.** ARC is a fundamental tool in studying the adversarial generalization of DNNs, since the adversarial generalization error bounds can be upper-bounded in terms of adequate upper-bounds for the ARC. Nevertheless, the appearance of the max operation in ARC's definition, complicates the derivation of upper bounds for it. Typical ways of circumventing this problem include the usage of optimal attacks or of a surrogate adversarial loss function [1, 12, 20, 50], employing a dual formulation of the maximization problem [35, 36], or working directly with the covering numbers of a DNN's adversarial hypothesis class [29, 49]. Similarly to [12, 49], we work with a surrogate FGSM-type loss and upper bound the ARC via the covering numbers of the DUN's adversarial hypothesis class. Nevertheless, we differentiate our approach by proving and using as a cornerstone the Lipschitz continuity of the examined DUN w.r.t. the parameter matrix. As depicted in Table 1, we derive a tighter upper-bound for the ARC, and thus for the adversarial generalization error, both in terms of attack level and number of layers (cf. Theorem 3 and Corollary 4), which is highly desirable. For a unified comparison, we consider the case of parameter matrices having upper-bounded spectral norms[1]. Our findings not only advance the theoretical understanding of DUNs under adversarial attacks, but also provide concrete tools for designing robust architectures for inverse problems, bridging a critical gap between empirical results and formal guarantees.

## 3 Background and problem formulation

**Notation.** For matrices $A_1, A_2 \in \mathbb{R}^{N \times N}$, we denote by $[A_1; A_2] \in \mathbb{R}^{2N \times N}$ their concatenation with respect to the first dimension, and $[A_1 | A_2] \in \mathbb{R}^{N \times 2N}$ their concatenation with respect to the second dimension. We write $O_{n \times n} \in \mathbb{R}^{n \times n}$ for the zero matrix and $I_{n \times n}$ for the $n \times n$ identity matrix. For $x \in \mathbb{R}, \tau > 0$, the soft thresholding operator $\mathcal{S}_\tau : \mathbb{R} \mapsto \mathbb{R}$ is defined as $\mathcal{S}_\tau(x) = \text{sign}(x) \max(0, |x| - \tau)$. For $x \in \mathbb{R}^n$, $\mathcal{S}_\tau(\cdot)$ acts component-wise and is 1-Lipschitz with respect to $x$. The covering number

---

[1]In the special case of low-rank parameter matrices, [12] exhibits a layer-independent adversarial generalization error bound, thereby rendering it tighter than ours in terms of the number of layers.

$\mathcal{N}(T, d, t)$ of a metric space $(T, d)$ at level $t > 0$, is defined as the smallest number of balls with respect to the metric $d$ required to cover $T$. When $d$ is induced by some norm $\|\cdot\|$, we write $\mathcal{N}(T, \|\cdot\|, t)$.

**An ADMM-based DUN for CS.** CS deals with recovering data $x \in \mathbb{R}^n$ from missing, noisy observations $y = Ax + e \in \mathbb{R}^m$, $m < n$, by assuming that there exists a *fixed* transform $W \in \mathbb{R}^{N \times n}$, $N \geq n$, so that $Wx \in \mathbb{R}^N$ is sparse. ADMM [6] is one of the most celebrated iterative algorithms solving the CS optimization problem: $\min_{x \in \mathbb{R}^n} \frac{1}{2} \|Ax - y\|_2^2 + \lambda \|Wx\|_1$, $\lambda > 0$. The output of ADMM at the $k$th iteration resembles the output of the $k$th layer of a DNN: it consists of a ReLU-type nonlinearity, i.e., the soft-thresholding operator, applied on an affine transformation of the input data. Then, unfolding ADMM relies on casting its iterations as layers of the said DNN.

To fully formulate ADMM as a trainable DNN, $W$ can be *unknown* and layer-dependent, so that it is learned from $\mathbf{S} = \{(x_i, y_i)\}_{i=1}^s \overset{\text{i.i.d.}}{\sim} \mathcal{D}^s$, for unknown $\mathcal{D}$. Overall, unfolding ADMM gives rise to the family of ADMM-based DUNs [25, 42], parameterized by $\{W_k\}_{k=1}^L$, for $L$ total layers. A prime representative from the family of ADMM-based DUNs is the state-of-the-art ADMM-DAD [25], which enjoys a sharing parameter property, i.e., $W = W_1 = \cdots = W_L$, thus allowing for less trainable parameters. To our knowledge, ADMM-DAD is the only ADMM-based DUN parameterized by an overcomplete sparsifier with $N > n$ — thereby leading to an overparameterized regime – with experimental results indicating a correlation between overcompleteness and robustness. This overcompleteness motivates us to set ADMM-DAD as a paradigm for studying the adversarial robustness and generalization of DUNs. The layer outputs of ADMM-DAD are given by

$$f^1(y) = I'b + I''\mathcal{S}_{\lambda/\rho}(b), \tag{1}$$

$$f^k(u) = I'(\Theta u + b) + I''\mathcal{S}_{\lambda/\rho}(\Theta u + b), \quad k = 2, \ldots, L, \tag{2}$$

for $\rho > 0$, $u^k \in \mathbb{R}^{2N \times 1}$, $\Theta = [I_{N \times N} - M \,|\, M] \in \mathbb{R}^{N \times 2N}$, $M := M_W = \rho W(A^T A + \rho W^T W)^{-1} W^T \in \mathbb{R}^{N \times N}$, $I' = [I_{N \times N}; O_{N \times N}] \in \mathbb{R}^{2N \times N}$, $I'' = [-I_{N \times N}; I_{N \times N}] \in \mathbb{R}^{2N \times N}$, $b := b_W(y) = W(A^T A + \rho W^T W)^{-1} A^T y \in \mathbb{R}^{N \times 1}$. With a slight abuse of notation, we write the composition of $L$ layer outputs as

$$f_W^L(y) = f^L \circ \cdots \circ f^1(y) \tag{3}$$

and call it the *intermediate decoder*. Then, ADMM-DAD implements the *final decoder*

$$h_W^L(y) = T_W(f_W^L(y)) = \hat{x} \approx x, \tag{4}$$

where $T_W(u) = \Lambda_W u + (A^T A + \rho W^T W)^{-1} A^T y \in \mathbb{R}^n$, with $\Lambda_W = [-\rho(A^T A + \rho W^T W)^{-1} W^T \,|\, \rho(A^T A + \rho W^T W)^{-1} W^T] \in \mathbb{R}^{n \times 2N}$. More details on ADMM and ADMM-DAD can be found at Appendix A.

**Definition 1** (**Parameter class of ADMM-DAD**). *We let $\mathcal{F}_\beta$ to be the class of all overcomplete sparsifiers $W \in \mathbb{R}^{N \times n}$ such that $S = W^T W$ is invertible and $\|S\|_{2 \to 2} \leq \beta$, for $0 < \beta < \infty$.*

**Remark 2.** *Due to the invertibility of $S$ [23], it holds $\alpha \leq \|S\|_{2 \to 2}$ and $\|W\|_{2 \to 2} \leq \sqrt{\beta}$, for some $0 < \alpha \leq \beta < \infty$.*

The standard hypothesis class [23] of all the decoders implemented by ADMM-DAD is $\mathcal{H}^L = \{h : \mathbb{R}^m \mapsto \mathbb{R}^n : h(y) = h_W^L(y), W \in \mathcal{F}_\beta\}$. Given $\mathcal{H}^L$ and the training dataset $\mathbf{S}$, ADMM-DAD aims to solve the CS problem by implementing $h_W(y) = \hat{x} \approx x$. We work towards that direction by minimizing (over $W$) the training mean-squared error (MSE): $\mathcal{L}_{\text{train}}(h) = \frac{1}{s} \sum_{i=1}^s \|h_W(y_i) - x_i\|_2^2$. Then, the generalization error – measuring the generalization performance of the network – is defined as $\text{GE}(h) = |\mathcal{L}_{\text{train}}(h) - \mathcal{L}_{\text{true}}(h)|$, with $\mathcal{L}_{\text{true}}(h) = \mathbb{E}_{(x,y) \sim \mathcal{D}}(\|h_W(y) - x\|_2^2)$ being the true error. Below, we give the counterparts for all errors under the adversarial learning setting.

**Adversarial learning and generalization.** In the presence of adversarial attacks $\delta$ during training and inference times, the *adversarial train MSE* and *adversarial true error* are given by $\widetilde{\mathcal{L}}_{\text{train}}(h) = \frac{1}{s} \sum_{i=1}^s \max_{\|\delta\|_p \leq \varepsilon} \|h_W(y_i + \delta_i) - x_i\|_2^2$ and $\widetilde{\mathcal{L}}_{\text{true}}(h) = \mathbb{E}_{(x,y) \sim \mathcal{D}}(\max_{\|\delta\|_p \leq \varepsilon} \|h_W(y + \delta) - x\|_2^2)$, respectively. Then, we aim to estimate the *adversarial generalization error*:

$$\widetilde{\text{GE}}(h) = |\widetilde{\mathcal{L}}_{\text{train}}(h) - \widetilde{\mathcal{L}}_{\text{true}}(h)|. \tag{5}$$

The appearance of the max operation poses extra difficulty in estimating $\widetilde{\text{GE}}$. To overcome this, a standard approach relies on considering adversarial attacks being the solution to the inner maximization problem. Similarly to [10], we rely on the FGSM, which is a so-called white-box attack, in the sense that the adversary has complete knowledge of the targeted model (and so $\delta$ depends on $W$), and we choose $p = 2$. Under this framework, FGSM yields $\delta := \delta_W^{\text{FGSM}} = \varepsilon \frac{\nabla_y \|h_W(y) - x\|_2^2}{\|\nabla_y \|h_W(y) - x\|_2^2\|_2}$, so that for

known $\delta$, we can discard the max operation in $\widetilde{\text{GE}}$ and rewrite it as $\widetilde{\text{GE}}(h) = |\widetilde{\mathcal{L}}_{\text{train}}(h) - \widetilde{\mathcal{L}}_{\text{true}}(h)| = |\frac{1}{s}\sum_{i=1}^{s} \|h_W(y_i + \delta_i^{\text{FGSM}}) - x_i\|_2^2 - \mathbb{E}_{(x,y)\sim\mathcal{D}}\|h_W(y + \delta^{\text{FGSM}}) - x\|_2^2|$. By attacking ADMM-DAD with $\delta$, we essentially perturb the input CS observations $y$, so the *perturbed intermediate and final decoders* are

$$f_W^L(y + \delta) = f^L \circ \cdots \circ f^1(y + \delta), \tag{6}$$

$$h_W^L(y + \delta) = T_W(f_W^L(y + \delta)), \tag{7}$$

respectively. The choice of $p = 2$ leads to a natural perturbation model, operating directly on the observations $y$ [32], and can provide a workable environment for studying adversarial generalization [10, 47]. Finally, we define the *adversarial hypothesis class* of ADMM-DAD as

$$\widetilde{\mathcal{H}}^L = \{\tilde{h} : \mathbb{R}^m \mapsto \mathbb{R}^n : \tilde{h}(y) = h_W^L(y + \delta) \mid h_W^L \in \mathcal{H}^L, \ W \in \mathcal{F}_\beta, \ \delta = \delta_W^{\text{FGSM}}\}. \tag{8}$$

Our goal is to study the adversarial generalization of ADMM-DAD, by delivering adversarial generalization error bounds over $\widetilde{\mathcal{H}}^L$. To do so, we employ the *adversarial Rademacher complexity*

$$\mathcal{R}_{\mathbf{S}}(\widetilde{\mathcal{H}}^L) = \mathbb{E}_\epsilon \sup_{\tilde{h}\in\widetilde{\mathcal{H}}^L} \frac{1}{s}\sum_{i=1}^{s} \epsilon_i \tilde{h}(y_i), \tag{9}$$

with $\epsilon$ being a vector with i.i.d. entries taking the values $\pm 1$ with equal probability. While prior approaches estimate (9) using covering numbers of an adversarial hypothesis class [49] and FGSM-based surrogate loss functions [12], our method introduces a distinct and principled refinement. We also bound the ARC of an adversarial hypothesis class parameterized by FGSM $l_2$-norm constrained attacks, deploying covering numbers. Nevertheless, our key innovation lies in establishing the Lipschitz continuity of the perturbed final decoder (7) with respect to the parameter matrix $W$. This structural property, which to our knowledge has not been previously exploited in this context, forms the foundation of our analysis and enables tighter ARC upper bounds (cf. Table 1) via the covering numbers of $\widetilde{\mathcal{H}}^L$. Due to the interpretability of unfolded architectures, we expect that similar results can be derived for other classes of adversarial attacks, under slight modifications. Below, we make a set of typical – in standard and adversarial learning scenarios – assumptions that will hold throughout the rest of the paper, and render our proofs and arguments relatively simple and accessible.

**Assumptions. ($\alpha$)** With high probability, we have $\|x_i\|_2 \leq B_{\text{in}}$, for some $B_{\text{in}} > 0$, $i = 1, \ldots, s$. For $\delta$ generated by the FGSM under an $l_2$-norm constraint, and for any $\tilde{h} \in \widetilde{\mathcal{H}}^L$, with high probability over $y_i$ chosen from $\mathcal{D}$, it holds $\|\tilde{h}(y_i)\|_2 \leq B_{\text{out}}$, for some $B_{\text{out}} > 0$, $i = 1, \ldots, s$. **(b)** For the soft-thresholding operator, we follow similar settings for nonsmooth functions [5] and write $\mathcal{S}'_{\lambda/\rho}(x) = 1$ for $|x| > \lambda/\rho$, and $\mathcal{S}'_{\lambda/\rho}(x) = 0$ for $|x| \leq \lambda/\rho$. **(c)** There exists some $\kappa > 0$ such that $\|\nabla_y \|f_W^k(Y) - X\|_2^2\|_2 \geq \kappa$, for any $W \in \mathcal{F}_\beta$, $k = 1, \ldots, L$. Boundedness assumptions for the gradient of the loss are standard when theoretically studying the adversarial robustness of DNNs [10, 12, 30], and are numerically supported [15, 33, 34], by imposing adequate constraints to avoid the case of $\|\nabla_y \|f_W^k(Y) - X\|_2^2\|_2 = 0$.

## 4 Main results

We address this paper's research question ($\star$), by delivering adversarial generalization error bounds for ADMM-DAD in the form of Theorem 3 (with proof found in Appendix C.5) and Corollary 4.

---

**Theorem 3 (Adversarial generalization error bounds for ADMM-DAD).** *For $L \geq 2$ being the total number of layers, let $\widetilde{\mathcal{H}}^L$ be the adversarial hypothesis class defined in (8) and $\delta$ adversarial attack generated by the FGSM, with $\|\delta\|_2 \leq \varepsilon$, for attack level $\varepsilon > 0$. Assume there exist pair-samples $\{(x_i, y_i)\}_{i=1}^s \overset{i.i.d.}{\sim} \mathcal{D}^s$, with $y_i = Ax_i + e$, $\|e\|_2 \leq \eta$, for some $\eta > 0$, and **Assumptions (a)** – **(c)** hold. Then with probability at least $1 - \zeta$, for all $\tilde{h} \in \widetilde{\mathcal{H}}^L$, the adversarial generalization error of ADMM-DAD defined in (5) is bounded as*

$$\widetilde{\text{GE}}(\tilde{h}) \leq O\left(\sqrt{\frac{Nn}{s}}\sqrt{\log\left(\exp\cdot\left(1 + \frac{2\sqrt{\beta}\text{Lip}_h^{L,\varepsilon}}{\sqrt{s}B_{\text{out}}}\right)\right)} + \sqrt{\frac{2\log(4/\zeta)}{s}}\right), \tag{10}$$

*with $\text{Lip}_h^{L,\varepsilon}$ – defined in Theorem 6 – being the Lipschitz constant of the adversarially perturbed final decoder (7) implemented by ADMM-DAD, and $\exp$ denoting the natural exponent.*

---

As we will see in Theorem 6, $L$ enters at most exponentially and $\varepsilon$ at most linearly in the definition of $\text{Lip}_h^{L,\varepsilon}$. Hence, due to the appearance of the logarithm in Theorem 3, we easily obtain:

**Corollary 4** (**Growth rate**). *If we consider the dependence of the adversarial generalization error bound* (10) *only on* $L, N, s, \varepsilon$, *and treat all other terms as constants, it roughly holds that*

$$\widetilde{\text{GE}}(\tilde{h}) \le O\left( \sqrt{\frac{NL \log(\varepsilon)}{s}} \right). \tag{11}$$

**Significance of Theorem 3 & Corollary 4.** *Our results: a) are **informative**, by including all elemental factors, i.e., $N, L, \varepsilon$, that determine the DUN's architecture and performance, b) **highlight** how overcompleteness $N$ ripples out to DUN adversarial generalization, i.e., although our bounds grow as $N$ increases, the growth is at the reasonable rate of $\sqrt{N}$, c) are **tighter** – which is desirable – and thus more realistic, than those of state-of-the-art related work [12, 49] w.r.t. $L$ and $\varepsilon$ (cf. Table 1).*

**The path to proving Theorem 3.** We give a sequence of results, each of which serves as a crucial component in deducing Theorem 3. We account for the number of training samples in **S** and thus pass to matrix notation, i.e., capitalize all vectors. Based on **Assumption (a)**, a simple application of the Cauchy-Schwartz inequality yields $\|Y\|_F \le \sqrt{s}B_{\text{in}}$, $\|\tilde{h}(Y)\|_F \le \sqrt{s}B_{\text{out}}$, and $\|\Delta\|_F \le \sqrt{s}\varepsilon$.

**Proposition 5** (**Bounded outputs**). *Let $k \in \mathbb{N}$, and $f_W^k(\cdot)$ be the perturbed intermediate decoder implemented by ADMM-DAD and defined in* (6). *Then, for any learnable overcomplete sparsifier $W \in \mathcal{F}_\beta$, we have*

$$\|f_W^k(Y + \Delta)\|_F \le (\|Y + \Delta\|_F)\|A\|_{2\to2}\nu\gamma\sqrt{\beta} \sum_{i=0}^{k-1} \nu^i(1 + 2\beta\gamma\rho)^i, \tag{12}$$

*where $\gamma = \frac{\rho}{\alpha - \rho\|A^T A\|_{2\to2}}$, $\alpha$ as in Remark 2, $\nu = 1 + \sqrt{2}$, and $\Delta := \Delta_W^{FGSM}$ with $\|\Delta_W^{FGSM}\|_F \le \sqrt{s}\varepsilon$.*

**Why is Proposition 5 important?** The upper bound on ADMM-DAD's outputs constitutes the first instance depicting how $L$ and $\varepsilon$ ripple out to the DUN adversarial robustness; then, we deploy this bound to prove Lipschitz continuity of $\tilde{h}_W^L(\cdot)$ with respect to $W$.

---

**Theorem 6** (**Lipschitz continuity of the perturbed decoder w.r.t. parameter**). *Let $\tilde{h}_W^L(\cdot)$ be the perturbed final decoder implemented by ADMM-DAD and defined in* (7), *$L \ge 2$, and learnable overcomplete sparsifier $W \in \mathcal{F}_\beta$. Then, for any $W_1, W_2 \in \mathcal{F}_\beta$, we have*

$$\|\tilde{h}_{W_1}^L(Y) - \tilde{h}_{W_2}^L(Y)\|_F := \|h_{W_1}^L(Y + \Delta_1) - h_{W_2}^L(Y + \Delta_2)\|_F \le \text{Lip}_h^{L,\varepsilon}\|W_1 - W_2\|_{2\to2}, \tag{13}$$

*where $\Delta_i := \Delta_{W_i}^{FGSM}$ with $\|\Delta_{W_i}^{FGSM}\|_F \le \sqrt{s}\varepsilon$, $i = 1, 2$, and Lipschitz constants $\text{Lip}_h^{L,\varepsilon}$ depending exponentially on the number of layers $L$ and linearly on the attack level $\varepsilon$:*

$$
\begin{aligned}
\text{Lip}_h^{L,\varepsilon} =& 2\gamma\rho \sqrt{\beta}\Bigg( (r\nu)^{L-1}\gamma\|A\|_{2\to2}\Big( r\|Y\|_F + r\sqrt{s}\varepsilon + 2\beta(B_{\text{in}} + B_{\text{out}})^2 \frac{\sqrt{s}\varepsilon}{\kappa^2}\nu\gamma^2\|A\|_{2\to2}^2 \Big) \\
&+ \sum_{k=2}^{L}(r\nu)^{L-k}H_k + \nu^2\gamma\|A\|_{2\to2}(\|Y\|_F + \sqrt{s}\varepsilon)\Big( 1 + \sum_{k=1}^{L-1}(r\nu)^k \Big) \Bigg),
\end{aligned}
\tag{14}
$$

*with $r = 1 + 2\beta\gamma\rho$, $\gamma$ as in Proposition 5, $\beta$ as in Definition 1, $\nu = (1 + \sqrt{2})$, and $H_k$ a constant calculated explicitly and defined in* (49) *of Appendix C.2.*

---

**Why is Theorem 6 important?** We provide an explicit formulation of $\text{Lip}_h^{L,\varepsilon}$, with direct dependence on $L$ and $\varepsilon$, allowing us to tightly upper-bound the ARC (9) with respect to $L$ and $\varepsilon$ (cf. Theorem 8).

While Theorem 6 enables tightness in the ARC bounds, the passage allowing the ARC's estimation is accomplished by means of the celebrated Dudley's inequality [9, Theorem 5.23], [11, Theorem 8.23]. This is a powerful probabilistic tool, which upper-bounds a stochastic process (like the ARC) on a space, to the integral of the covering numbers of this space. We work towards that direction and define $\widetilde{\mathcal{M}} := \{(\tilde{h}(y_1)|\dots|\tilde{h}(y_s)) \in \mathbb{R}^{n\times s} : \tilde{h} \in \widetilde{\mathcal{H}}^L\} = \{(h_W^L(y_1 + \delta_1)|\dots|h_W^L(y_s + \delta_s)) \in \mathbb{R}^{n\times s} : h_W = h \in \mathcal{H}^L, W \in \mathcal{F}_\beta\}$, corresponding to $\widetilde{\mathcal{H}}^L$. Since $\widetilde{\mathcal{M}}$ and $\widetilde{\mathcal{H}}^L$ are parameterized by $W$, we rewrite (9) as

$$\mathcal{R}_{\mathbf{S}}(\widetilde{\mathcal{H}}^L) = \mathbb{E} \sup_{\tilde{h} \in \widetilde{\mathcal{H}}^L} \sum_{i=1}^{s} \sum_{k=1}^{n} \epsilon_{ik}\tilde{h}_k(y_i) = \mathbb{E} \sup_{M \in \widetilde{\mathcal{M}}} \frac{1}{s} \sum_{i=1}^{s} \sum_{k=1}^{n} \epsilon_{ik}M_{ik}, \tag{15}$$

so that we can estimate the covering numbers of $\widetilde{\mathcal{M}}$ instead of $\widetilde{\mathcal{H}}^L$:

**Proposition 7** (**Upper-bound on covering numbers**). *For the covering numbers of $\widetilde{\mathcal{M}}$ it holds:*

$$\mathcal{N}(\widetilde{\mathcal{M}}, \|\cdot\|_F, t) \le \left(1 + \frac{2\sqrt{\beta}\mathrm{Lip}_h^{L,\varepsilon}}{t}\right)^{Nn}, \tag{16}$$

*with $\mathrm{Lip}_h^{L,\varepsilon}$ defined as in Theorem 6.*

Thanks to (16), overcompleteness $N$ is also included in the framework we are setting up. Then, a simple application of the Dudley's integral inequality upper-bounds the ARC in terms of (16):

> **Theorem 8** (**ARC estimate**). *Let $\widetilde{\mathcal{H}}^L$ be the adversarial hypothesis class of ADMM-DAD and defined in (8). Then, for the adversarial Rademacher complexity $\mathcal{R}_\mathbf{S}(\widetilde{\mathcal{H}}^L)$ defined in (9) it holds:*
>
> $$\mathcal{R}_\mathbf{S}(\widetilde{\mathcal{H}}^L) \le \frac{4\sqrt{2}}{s} \int_0^{\frac{\sqrt{s}B_{\mathrm{out}}}{2}} \sqrt{Nn \log\left(1 + \frac{2\sqrt{\beta}\mathrm{Lip}_h^{L,\varepsilon}}{t}\right)} dt. \tag{17}$$

**Why is Theorem 8 important?** The ARC is an essential tool for thoroughly explaining adversarial generalization. The explicit dependence of the ARC estimate on elemental quantities like $N$, $L$, $\varepsilon$, stresses how these ripple out to the adversarial generalization of ADMM-DAD. Especially, by definition of $\mathrm{Lip}_h^{L,\varepsilon}$, due to the appearance of the logarithm, and since (17) can be proven to be integral-free (cf. Appendix C.5), the ARC estimate roughly scales like $\sqrt{NL\log(\varepsilon)}$ (cf. Corollary 4).

To connect the ARC to the adversarial generalization error bound and deduce Theorem 3, we deploy [38, Theorem 26.5]. The latter upper-bounds the generalization error of a network, to the Rademacher complexity of the network's hypothesis class, when composed with the loss $\|\cdot\|_2^2$. To remove $\|\cdot\|_2^2$ and work solely with the Rademacher complexity, we employ [28, Corollary 4], which further requires to calculate the Lipschitz constant of $\|\cdot\|_2^2$. It is easy to check that $\ell(\cdot) = \|\cdot\|_2^2$ is Lipschitz continuous, with Lipschitz constant $\mathrm{Lip}_{\|\cdot\|_2^2} = 2B_{\mathrm{in}} + 2B_{\mathrm{out}}$. Therefore, by (17) and [28, Corollary 4] we deduce:

$$\mathcal{R}_\mathbf{S}(\|\cdot\|_2^2 \circ \widetilde{\mathcal{H}}^L) \le \sqrt{2}(2B_{\mathrm{in}} + 2B_{\mathrm{out}})\mathcal{R}_\mathbf{S}(\widetilde{\mathcal{H}}^L) \le O\left(\int_0^{\frac{\sqrt{s}B_{\mathrm{out}}}{2}} \sqrt{Nn \log\left(1 + \frac{2\sqrt{\beta}\mathrm{Lip}_h^{L,\varepsilon}}{t}\right)} dt\right). \tag{18}$$

We combine (18) and [38, Theorem 26.5], to give adversarial generalization error bounds for ADMM-DAD, stated formally in Theorem 3 and Corollary 4, thus answering this paper's research question ($\star$). The proofs of Proposition 5, Theorem 6, Proposition 7 and Theorem 8 can be found at Appendices C.1, C.2, C.3 and C.4, respectively. Moreover, [11, Theorem 8.23], [38, Theorem 26.5] and [28, Corollary 4] are formally stated as Theorem B.6, Theorem B.7 and Lemma B.8, respectively. In the next Section, we assess the validity of our theory with a series of experiments on real-world data.

## 5 Experiments

We train and test ADMM-DAD on two real-world image datasets: CIFAR10 (50000 training and 10000 test $32 \times 32$ coloured image examples) and SVHN (73257 training and 26032 test $32 \times 32$ colored image examples). For both datasets, we transform the images into grayscale ones and vectorize them. We fix $m/n = 25\%$, and alternate the overcompleteness $N$, and the number of layers $L$. We consider a standard CS setup, with an appropriately normalized random Gaussian $A \in \mathbb{R}^{m \times n}$, and noisy observations of the form $y = Ax + e$, with $e$ being zero-mean Gaussian noise, with standard deviation $\mathrm{std} = 10^{-2}$. To generate the adversarial attack $\delta$, we employ FGSM from [15], under an $l_2$-norm constraint, and attack ADMM-DAD with $\delta$ during training and test times, with varying attack levels $\varepsilon$. To highlight the adversarial robustness of ADMM-DAD against more powerful attacks, we also employ [15] to generate an $\ell_2$-based PGD attack with 10 iterations. We initialize the learnable overcomplete sparsifier $W \in \mathbb{R}^{N \times n}$ using a Xavier normal distribution [14]. We implement all models in PyTorch [19] and train them using the Adam algorithm [21], with batch size 128. As evaluation metrics, we use the *clean and adversarial test MSEs*

$$\mathcal{L}_{\mathrm{test}}(h) = \frac{1}{d} \sum_{i=1}^d \|h(y_i') - x_i'\|_2^2, \tag{19}$$

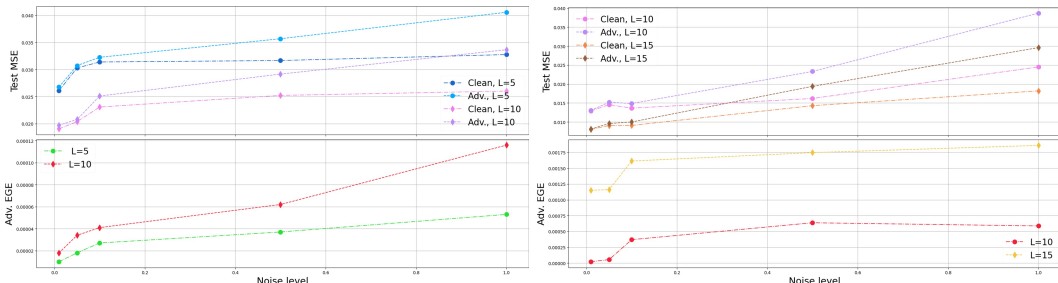

**Figure 1:** Performance of ADMM-DAD (plotted on logarithmic scale) on CIFAR10 (left) and SVHN (right), for varying number of layers $L$ and attack levels $\varepsilon$ of the FGSM, and overcompleteness $N = 10n$. Top: clean test MSE (19) and adversarial test MSE (20). Bottom: adversarial EGE (21). For both datasets, (21) increases as Theorem 3 suggests, and in fact scales at the rate dictated by Corollary 4, thus confirming our derived generalization theory. A similar increment is observed for both (19) and (20), but at a reasonable rate, thereby highlighting the adversarial robustness of ADMM-DAD.

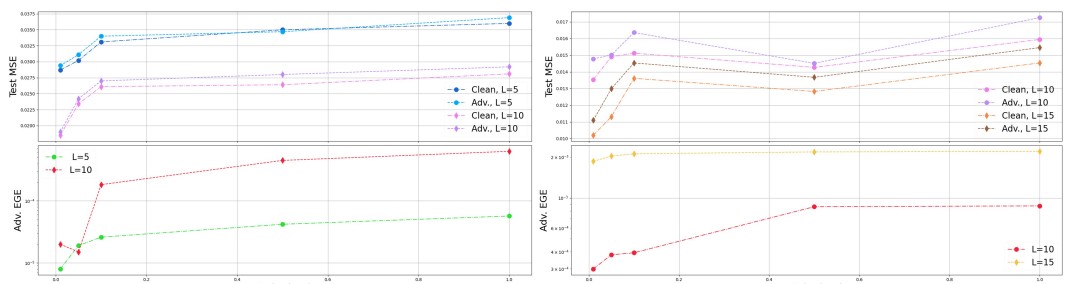

**Figure 2:** Performance of ADMM-DAD (plotted on logarithmic scale) on CIFAR10 (left) and SVHN (right), for varying number of layers $L$ and attack levels $\varepsilon$ of the PGD (10 iterations), and overcompleteness $N = 10n$. Top: clean test MSE (19) and adversarial test MSE (20). Bottom: adversarial EGE (21). Although our theoretical analysis focuses on FGSM, we observe that even for a stronger adversarial attack like PGD, (21) scales at the rate dictated by Corollary 4, for both datasets, thus corroborating our derived generalization theory. Similarly, (19) and (20) also increase, but at a reasonable rate, thus highlighting the adversarial robustness of ADMM-DAD, even under more powerful than FGSM attacks.

$$\widetilde{\mathcal{L}}_{\text{test}}(h) = \frac{1}{d} \sum_{i=1}^{d} \|h(y_i' + \delta_i') - x_i'\|_2^2, \tag{20}$$

respectively, with $\mathbf{D} = \{(y_i', x_i')\}_{i=1}^{d}$ being a set of $d$ test data, not used during training, and the *adversarial empirical generalization error* (adversarial EGE)

$$\widetilde{\text{EGE}}(h) = |\widetilde{\mathcal{L}}_{\text{test}}(h) - \widetilde{\mathcal{L}}_{\text{train}}(h)|, \tag{21}$$

with $\widetilde{\mathcal{L}}_{\text{train}}(h)$ being the adversarial train MSE defined in Sec. 3. We also compare ADMM-DAD to a state-of-the-art baseline DUN called ISTA-net [4], parameterized by an orthogonal sparsifier. We aim to use the structural difference between the two DUNs to showcase how the adversarial robustness and generalization of DUNs are affected, when employing an overcomplete sparsifier instead of an orthogonal one. For more experimental settings and details, we refer the reader to Appendix D.

**Test and generalization errors for increasing $\varepsilon$.** We measure the performance of ADMM-DAD with $L = 5$ and $L = 10$ on CIFAR10, and $L = 10$ and $L = 15$ on SVHN, both with fixed $N = 10n$, in terms of the clean test MSE (19), the adversarial test MSE (20), and the adversarial EGE (21), as $\varepsilon$ varies, for both the FGSM and the PGD. We report the results in Figure 1 for FGSM, and in Figure 2 for PGD, both corroborating Theorem 3. Specifically, for both datasets and attacking methods, we observe that the adversarial EGEs, as these are depicted in the bottom of Figure 1 and Figure 2, increase, as both $L$ and $\varepsilon$ increase. Despite the appearance of other terms in (10), the adversarial EGEs seem to scale at the rate of $\sqrt{L\log(\varepsilon)}$, like Corollary 4 suggests. As illustrated at the top of Figure 1 and Figure 2, both (19) and (20) also increase as $\varepsilon$ increases. This behavior is anticipated from the adversarial robustness perspective, since the higher the attack level, the more a neural network "struggles" to infer correctly. Nevertheless, the increment of (19) and (20) on both

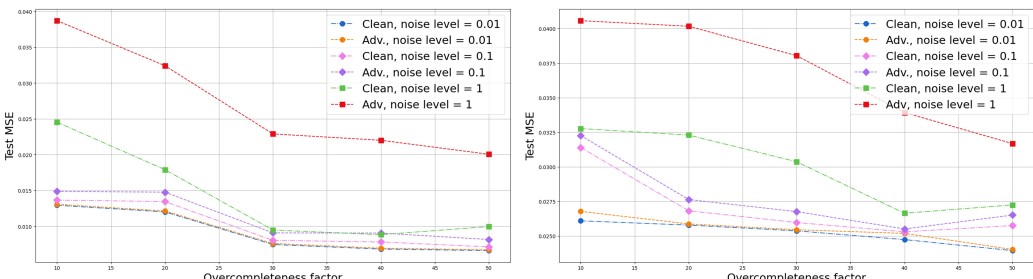

**Figure 3:** Robustness plots of 5-layer ADMM-DAD on CIFAR10, (left) and 10-layer ADMM-DAD on SVHN (right), for alternating overcompleteness $N$ and different attack levels $\varepsilon$ of FGSM. For both datasets, as $N$ increases, the clean test MSEs (19) and the adversarial test MSEs (20) drop. Importantly, for the standard case of $\varepsilon = 0.01$, the robustness gap of ADMM-DAD on both datasets is particularly small. All in all, results highlight the beneficial role that $N$ plays on robustifying ADMM-DAD against varying adversarial attack levels.

datasets seem to be at the reasonable rate of roughly square-root. Overall, the empirical behavior of ADMM-DAD matches its theoretical one. Interestingly, despite the fact that our theory hinges upon the FGSM, the similar adversarial robustness and generalization of the PGD-attacked ADMM-DAD fuels us to theoretically investigate this phenomenon more in the future, as we pinpoint in Sec. 6.

**The role of $N$ in adversarial robustness.** We measure the adversarial robustness of 5- and 10-layer ADMM-DAD on CIFAR10 and SVHN, respectively, for increasing $N$, and three different values of $\varepsilon$ for the FGSM, by means of (19) and (20). Given that DUNs operate in a regression setting (see (7)) – so that standard classification-type metrics like accuracy cannot be utilized – the deployment of the clean and adversarial test MSEs (and their gap) as adversarial robustness metrics is a common practice [35, 36]. We report the results for both datasets in Figure 3, which demonstrates an intriguing phenomenon: as $N$ increases, for a fixed $\varepsilon$, both clean and adversarial test MSEs drop, indicating that the overcompleteness of the learnable sparsifying transform promotes ADMM-DAD's adversarial robustness. Of course, the MSEs increase as $\varepsilon$ also increases, but this is anticipated, since a stronger attack in the CS observations tantamounts to a DUN being less able to recover the data. Nevertheless, explaining the adversarial robustness of ADMM-DAD through the overcompleteness of the learnable sparsifier lays a fruitful ground, to identify "hyper-parameters" and properly fine-tune them, so as to boost adversarial robustness in the unfolding regime; we briefly mention this research line in Sec. 6.

**A note on the robustness gap.** Based on Figures 1 and 3, we examine the robustness gap of ADMM-DAD, i.e., the difference between (19) and (20). We notice that the robustness gap slightly increases for the CIFAR10 with increasing $\varepsilon$ of the FGSM – albeit the gap's scaling from one attack level to the next is reasonable – but the picture, e.g., Figure 1, is much better for the SVHN, where both test errors grow proportionally. This indicates that ADMM-DAD enjoys adversarial robustness to increasing attack levels. Especially in the case of $\varepsilon = 0.01$ – which is still a non-negligible attack level – we deduce from Figure 3 that, for both datasets, the robustness gap is especially small, e.g., $\sim 10^{-4}$. Due to these interesting findings, explaining the robustness gap of DUNs, both from a theoretical and practical side, could be an inspiring future work (see also Sec. 6).

**Comparison to baseline.** We compare a 10-layer ADMM-DAD to a 10-layer ISTA-net on both datasets, with a mild overcompleteness for ADMM-DAD of $N = 10n$. We present the results in Table 3, with the top comparisons refering to CIFAR10 and the bottom to SVHN. We observe that ADMM-DAD outperforms the baseline, consistently for all datasets, since it exhibits smaller clean and adversarial test MSEs, as well as adversarial EGE; on the other hand, ISTA-net achieves errors being orders of magnitude larger than those of ADMM-DAD. Furthermore, we observe that the robustness gap (see paragraph above) exhibited by ADMM-DAD is smaller than the corresponding one of ISTA-net. This behavior highlights the beneficial role of overcompleteness in the unfolding regime, as opposed to orthogonality, when studying adversarial robustness.

## 6 Conclusion and future work

In this paper, we addressed the adversarial generalization of DUNs. These are interpretable networks emerging from iterative optimization algorithms, incorporate knowledge of the data model, and

**Table 2:** Comparison of ADMM-DAD – with overcompleteness $N = 10n$ – to the baseline, both with 10 layers, against different FGSM attack levels $\varepsilon$, on CIFAR10 (top) and SVHN (bottom). Bold letters indicate the DUN that scores the best performance in terms of all metrics (19), (20), (21). Overall, ADMM-DAD outperforms the baseline, highlighting the advantage of overcompleteness over orthogonality in the unfolding regime.

| CIFAR10 | Clean test MSE | | | Adv. test MSE | | | Adv. EGE | | |
|---|---|---|---|---|---|---|---|---|---|
| DUN $\quad\varepsilon$ | 0.01 | 0.1 | 1 | 0.01 | 0.1 | 1 | 0.01 | 0.1 | 1 |
| **ADMM-DAD** | 0.019 | 0.023 | 0.026 | 0.020 | 0.025 | 0.034 | $0.18 \cdot 10^{-4}$ | $0.41 \cdot 10^{-4}$ | $1.16 \cdot 10^{-4}$ |
| ISTA-net | 0.021 | 0.027 | 0.229 | 0.023 | 0.047 | 0.229 | $0.70 \cdot 10^{-2}$ | $0.55 \cdot 10^{-2}$ | $0.15 \cdot 10^{-2}$ |
| SVHN | Clean test MSE | | | Adv. test MSE | | | Adv. EGE | | |
| DUN $\quad\varepsilon$ | 0.01 | 0.1 | 1 | 0.01 | 0.1 | 1 | 0.01 | 0.1 | 1 |
| **ADMM-DAD** | 0.013 | 0.014 | 0.025 | 0.013 | 0.015 | 0.039 | $0.22 \cdot 10^{-4}$ | $3.77 \cdot 10^{-4}$ | $5.87 \cdot 10^{-4}$ |
| ISTA-net | 0.028 | 0.036 | 0.676 | 0.030 | 0.054 | 0.703 | $0.89 \cdot 10^{-2}$ | $0.78 \cdot 10^{-2}$ | $15 \cdot 10^{-2}$ |

are designed to solve inverse problems like CS, which finds applications in safety-related domains. Thus, it is crucial to understand the adversarial robustness and generalization of DUNs. To that end, we selected an overparameterized representative from a celebrated family of DUNs, serving as a paradigm to study the adversarial generalization in the unfolding regime; then, we perturbed this network with FGSM-based attacks under an $l_2$-norm constraint. We proved that the attacked network is Lipschitz continuous with respect to the parameters – a crucial intermediate step for estimating the DUN's adversarial Rademacher complexity. Then, we utilized this estimate to deliver adversarial generalization error bounds for the representative DUN. To our knowledge, these are the first theoretical results explaining the adversarial generalization of DUNs. Finally, we supported our theory with relevant experiments, and highlighted how overparameterization in the unfolding regime can promote adversarial robustness.

Our work opens promising future directions. Although we provided a solid mathematical explanation for the adversarial generalization in the unfolding regime, a question arises regarding the tightness of these upper bounds, for instance, with respect to the overparameterization. What is more, the generalization of our theoretical framework to broader classes of adversarial attacks like PGD-based with different norm constraints, could improve the understanding and impact of DUNs in real-world scenarios, e.g., when these are applied in CS-MRI. Finally, we empirically observed that overparameterization promotes adversarial robustness in the unfolding regime. Consequently, it would be fruitful to theoretically study the robustness gap, in terms of the overparameterization, as a means of explaining the adversarial robustness of DUNs.

## Acknowledgments and Disclosure of Funding

The research was mainly conducted during the author's affiliation with the Isaac Newton Institute for Mathematical Sciences of the University of Cambridge. The author would like to thank the Isaac Newton Institute for Mathematical Sciences for supporting them during their INI Postdoctoral Research Fellowship in the Mathematical Sciences, especially during the program "Representing, calibrating & leveraging prediction uncertainty from statistics to machine learning". This work was funded by the EPSRC (Grant Number EP/V521929/1). Moreover, the author would like to thank P. Mertikopoulos for insightful comments and fruitful discussions.

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

# Appendices

## A   Unfolding ADMM to ADMM-DAD for solving Compressed Sensing

In this Section, we introduce the inverse problem of Compressed Sensing (CS), and present two methods to solve it: the model-based iterative algorithm ADMM and its unfolded counterpart, namely, ADMM-DAD. For the sake of completeness, we restate parts of the background presented in Sec. 3.

### A.1   Preliminaries on Compressed Sensing

CS deals with recovering data $x \in \mathbb{R}^n$ from missing, noisy observations $y = Ax + e \in \mathbb{R}^m$, $m < n$, $\|e\|_2 \leq \eta$, $\eta > 0$, under the assumption that there exists some *fixed* sparsifying transform $W \in \mathbb{R}^{N \times n}$, $N \geq n$. In a typical CS scenario, for additive Gaussian noise $e$ of sufficiently high standard deviation, e.g., of the order of $10^{-2}$, we have approximate recovery of $x$, so that the error between the original and reconstructed data is upper-bounded by a quantity involving $\eta$.

Under the sparsity assumption, the optimization problem associated to CS is the so-called LASSO

$$\min_{x \in \mathbb{R}^n} \frac{1}{2} \|Ax - y\|_2^2 + \lambda \|Wx\|_1.$$

In the usual *synthesis sparsity* model, $W$ is usually an orthogonal sparsifying transforms, i.e., $W \in \mathbb{R}^{n \times n}$, with $WW^T = I_{n \times n}$ (e.g. $W$ may be the discrete cosine transform). However, when $W$ is *overcomplete*, namely, $N > n$, one operates in the much more flexible *analysis sparsity* model, which is shown to offer more advantages than its synthesis counterpart. For an analytic comparison between the two sparsity models, we refer the interested reader to [23, 24]. Under the analysis sparsity model, the optimization problem of CS is called *generalized* LASSO.

### A.2   Model-based approach: the ADMM

Various algorithms can solve the (generalized) LASSO problem, one of which is the celebrated *alternating direction method of multipliers* (ADMM). ADMM introduces dual variables $z, v \in \mathbb{R}^N$, so that the LASSO problem is equivalent to

$$\min_{x \in \mathbb{R}^n} \frac{1}{2} \|Ax - y\|_2^2 + \lambda \|z\|_1 \quad \text{subject to} \quad Wx - z = 0.$$

For $\rho > 0$ (penalty parameter), $k \in \mathbb{N}$ iterations, initial points $(x^0, z^0, v^0)$, and $\mathcal{S}_{\lambda/\rho}(\cdot)$ being the soft-thresholding operator introduced in Sec. 3, ADMM produces the following iterative scheme:

$$x^{k+1} = (A^T A + \rho W^T W)^{-1}(A^T y + \rho W^T(z^k - v^k))$$
$$z^{k+1} = \mathcal{S}_{\lambda/\rho}(Wx^{k+1} + v^k)$$
$$v^{k+1} = Wx^{k+1} + v^k - z^{k+1},$$

which is known [6] to converge to a solution $p^\star$ of the (generalized) LASSO's equivalent formulation, i.e., $\|Ax^k - y\|_2^2 + \|z^k\|_1 \to p^\star$ and $Wx^k - z^k \to 0$ as $k \to \infty$. Although ADMM can equally address the LASSO problem under both the synthesis and the analysis sparsity models, we continue our setup with an overcomplete sparsifier $W \in \mathbb{R}^{N \times n}$, due to the advantages of analysis over synthesis sparsity.

### A.3   Data-driven approach: the ADMM-DAD

To reformulate the iterative scheme of ADMM as a deep unfolding network (DUN), we substitute the $x$-update into the $z$- and $v$-updates, then the $z$-update into the $v$-update, and introduce the *intermediate variable* $u_k = [v^k; z^k] \in \mathbb{R}^{2N \times 1}$, so that

$$v^k = \Theta u^k + b - \mathcal{S}_{\lambda/\rho}(\Theta u^k + b)$$
$$z^k = O_{N \times 2N} u^k + O_{N \times N} b + \mathcal{S}_{\lambda/\rho}(\Theta u^k + b),$$

with

$$\Theta = [I_{N \times N} - M \,|\, M] \in \mathbb{R}^{N \times 2N}$$

$$M := M_W = \rho W (A^T A + \rho W^T W)^{-1} W^T \in \mathbb{R}^{N \times N}$$
$$b := b_W(y) = W(A^T A + \rho W^T W)^{-1} A^T y \in \mathbb{R}^{N \times 1}.$$

Finally, we introduce $I' = [I_{N \times N}; O_{N \times N}] \in \mathbb{R}^{2N \times N}$ and $I'' = [-I_{N \times N}; I_{N \times N}] \in \mathbb{R}^{2N \times N}$, so that the iterative scheme of ADMM is compactly written in the following single-variable form:

$$u^{k+1} = I'(\Theta u^k + b) + I'' \mathcal{S}_{\lambda/\rho}(\Theta u^k + b), \qquad k \geq 0.$$

To enable a learning scenario, we assume that the sparsifying transform $W$ is *unknown and learned* from a set of i.i.d. training samples, i.e., $\mathbf{S} = \{(x_i, y_i)\}_{i=1}^{s}$, drawn from an unknown distribution[2] $\mathcal{D}^s$. Then, the iterative scheme of ADMM can be interpreted as a neural network with $L \in \mathbb{N}$ layers, coined ADMM Deep Analysis Decoding (ADMM-DAD) [25].

# B   Auxiliary Theorems

In this Section, we state Theorems, which will be used later on in the proofs of our main results. Specifically, we start with two well-known Theorems from numerical linear algebra.

**Theorem B.1.** *Let $A \in \mathbb{R}^{n \times n}$ be invertible and $B \in \mathbb{R}^{n \times n}$. For a sub-multiplicative matrix norm $\|\cdot\|$ on $\mathbb{R}^{n \times n}$, if it holds $\|A^{-1}\| \cdot \|B\| < 1$, then $A + B \in \mathbb{R}^{n \times n}$ is invertible. Moreover, we have*

$$\|(A+B)^{-1}\| \leq \frac{\|A^{-1}\|}{1 - \|A^{-1}\| \cdot \|B\|}. \tag{22}$$

*Proof.* If $A + B$ is not invertible, then there exists some $x \neq 0$ such that $Ax + Bx = 0$. By assumption, $A$ is invertible, thus $-x = A^{-1}Bx$. Hence, $\|x\| = \|A^{-1}Bx\| \leq \|A^{-1}\| \cdot \|B\| \cdot \|x\| \overset{x \neq 0}{\implies} 1 \leq \|A^{-1}\| \cdot \|B\|$, which contradicts our assumption, so $A + B$ is invertible. We also have: $A^{-1}(A+B) = I - (-A^{-1}B) \implies A + B = A(I + A^{-1}B)$. Since $A + B$ and $I + A^{-1}B$ are invertible, we get $(A+B)^{-1} = A^{-1}(I + A^{-1}B)^{-1}$. Due to the invertibility of $I + A^{-1}B$, we get

$$(I + A^{-1}B)^{-1} + A^{-1}B(I + A^{-1}B)^{-1} = I$$
$$\iff (I + A^{-1}B)^{-1} = I - A^{-1}B(I + A^{-1}B)^{-1}$$
$$\iff \|(I + A^{-1}B)^{-1}\| = \|I - A^{-1}B(I + A^{-1}B)^{-1}\|$$
$$\leq \|I\| + \|A^{-1}B(I + A^{-1}B)^{-1}\|$$
$$\leq 1 + \|A^{-1}\| \cdot \|B\| \cdot \|(I + A^{-1}B)^{-1}\|$$
$$\implies \|(I + A^{-1}B)^{-1}\| \leq \frac{1}{1 - \|A^{-1}\| \cdot \|B\|}.$$

We apply the latter estimate to $\|(A+B)^{-1}\| \leq \|A^{-1}\| \cdot \|(I + A^{-1}B)^{-1}\|$ and the proof follows. $\quad\square$

**Theorem B.2.** *For a sub-multiplicative matrix norm $\|\cdot\|$ on $\mathbb{R}^{n \times n}$, if $A$, $B \in \mathbb{R}^{n \times n}$ are invertible, then*

$$\|B^{-1} - A^{-1}\| \leq \|B^{-1}\| \cdot \|A^{-1}\| \cdot \|A - B\|. \tag{23}$$

*Proof.* Since $B^{-1} - A^{-1} = B^{-1}(I - BA^{-1}) = B^{-1}(AA^{-1} - BA^{-1}) = B^{-1}(A - B)A^{-1}$, we deduce, by sub-multiplicativity of the norm $\|\cdot\|$, that $\|B^{-1} - A^{-1}\| \leq \|B^{-1}\| \cdot \|(A - B)\| \cdot \|A^{-1}\|$. $\quad\square$

As part of our strategy for proving in Appendix C.2 the Lipschitz continuity of the perturbed final decoder (7), we provide below two intermediate results, which serve in a similar way to Proposition 5: a) in Proposition B.3 we show that the gradient – with respect to the input – of the layer outputs, is upper-bounded by a quantity involving the number of layers $k < L$, b) Theorem B.4 showcases the Lipschitz continuity of the final decoder (4) – in the clean, not contaminated by adversarial attacks regime – with respect to the parameter matrix $W$. In fact, the Lipschitz constants of the decoder depend exponentially on the total number of layers $L$.

---

[2]Formally speaking, this is a distribution over $x_i$ and for fixed $A, e$, we obtain $y_i = Ax_i + e$

**Proposition B.3** (Bounded gradient outputs). *Let $k \in \mathbb{N}$, and $f_W^k(\cdot)$ be the intermediate decoder of ADMM-DAD defined in (3). Then, for any learnable overcomplete sparsifier $W \in \mathcal{F}_\beta$, we have*

$$\|\nabla_Y f_W^k(Y)\|_F \leq \|A\|_{2\to 2} \nu\gamma \sqrt{\beta} \sum_{i=0}^{k-1} \nu^i (1 + 2\beta\gamma\rho)^i, \tag{24}$$

*where $\nu = 1 + \sqrt{2}$, $\gamma = \frac{\rho}{\alpha - \rho\|A^T A\|_{2\to 2}}$, $\beta$ as in Definition 1 and $\alpha$ as in Remark 2.*

*Proof.* We prove (24) via induction. Firstly, we notice that $\|I' + I''\|_{2\to 2} \leq \|I'\|_{2\to 2} + \|I''\|_{2\to 2} \leq 1 + \sqrt{2} = \nu$. Then, for $k = 1$:

$$\|\nabla_Y f_W^1(Y)\|_F \leq \nu\|\nabla_Y B\|_F \leq \nu\|A\|_{2\to 2} \sqrt{\beta}\|(A^T A + \rho W^T W)^{-1}\|_{2\to 2},$$

which holds by definition of (1). The invertibility of $S = W^T W$ and Theorem B.1 imply that

$$\|(A^T A + \rho W^T W)^{-1}\|_{2\to 2} = \|(A^T A + \rho S)^{-1}\|_{2\to 2} \leq \frac{\rho\|S^{-1}\|_{2\to 2}}{1 - \rho\|S^{-1}\|_{2\to 2}\|A^T A\|_{2\to 2}}$$

$$= \frac{\rho}{\alpha - \rho\|A^T A\|_{2\to 2}} := \gamma,$$

where in the last inequality we used the fact that $\beta^{-1} \leq \|S^{-1}\|_{2\to 2} \leq \alpha^{-1}$, due to the overcompleteness of $W$ [23]. Hence,

$$\|\nabla_Y f_W^1(Y)\|_F \leq \|A\|_{2\to 2} \nu\gamma \sqrt{\beta}.$$

Suppose now that (24) holds for some $k \in \mathbb{N}$. Then, for $k + 1$:

$$\begin{aligned}
\|\nabla_Y f_W^{k+1}(Y)\|_F &\leq \nu(\|\Theta\|_{2\to 2}\|\nabla_Y f_W^k(Y)\|_F + \|\nabla_Y B\|_F) \\
&\leq \nu\left((1 + 2\|M\|_{2\to 2})\|\nabla_Y f_W^k(Y)\|_F + \|\nabla_Y B\|_F\right) \\
&\leq \nu\left((1 + 2\beta\gamma\rho)\left(\nu\|A\|_{2\to 2}\gamma \sqrt{\beta} \sum_{i=0}^{k-1} \nu^i(1 + 2\beta\gamma\rho)^i\right) + \|A\|_{2\to 2}\gamma \sqrt{\beta}\right) \\
&= \|A\|_{2\to 2}\nu\gamma \sqrt{\beta} \sum_{i=0}^{k} \nu^i(1 + 2\beta\gamma\rho)^i,
\end{aligned}$$

which complements the proof. $\qquad\square$

**Theorem B.4** (Lipschitz continuity of final decoder w.r.t. parameter – [23, Corollary 3.11]). *Let $h \in \mathcal{H}^L$ be the standard hypothesis class of ADMM-DAD:*

$$\mathcal{H}^L = \{h : \mathbb{R}^m \mapsto \mathbb{R}^n : h(y) = h_W^L(y), \ W \in \mathcal{F}_\beta\},$$

*$L \geq 2$ be the total number of layers, and learnable overcomplete sparsifier $W \in \mathcal{F}_\beta$, with $\mathcal{F}_\beta$ as in Definition 1. Then, for any $W_1, W_2 \in \mathcal{F}_\beta$, we have:*

$$\|T_{W_2}(f_{W_2}^L(Y)) - T_{W_1}(f_{W_1}^L(Y))\|_F \leq \Sigma_L\|W_2 - W_1\|_{2\to 2}, \tag{25}$$

*where*

$$\Sigma_L = 2\gamma\rho \sqrt{\beta}\left(K_L + \nu\gamma\|A\|_{2\to 2}\|Y\|_F(1 + 2\beta\gamma\rho) \sum_{k=0}^{L-1} \nu^k(1 + 2\beta\gamma\rho)^k\right), \tag{26}$$

*with $\nu = 1 + \sqrt{2}$, $\gamma$ as in Proposition 5, $\beta$ as in Definition 1, and*

$$K_L = \gamma G^L + \sum_{k=2}^{L} \left( G^{L-k} \left[ \gamma G + 4G\beta\gamma^2 \rho \|A\|_{2\to2} \|Y\|_F \sum_{i=0}^{k-2} G^i \right] \right), \tag{27}$$

*with $G = \nu(1 + 2\beta\gamma\rho)$.*

Two key concepts on which the estimation of the (adversarial) Rademacher complexity relies are the covering numbers and Dudley's integral inequality. Specifically, in order to estimate the supremum of a stochastic process – which is essentially how the ARC is defined – over a space, one can employ Dudley's inequality [9, Theorem 5.23], [11, Theorem 8.23], which further requires the estimation of the covering numbers of the said space.

To that end, we state below an intermediate result, which provides an upper-bound on the covering numbers of the $\alpha$-radius ball in the space of all $N \times n$ real-valued matrices. From this space, we pass to the estimation of the covering numbers of the parameter space $\mathcal{F}_\beta$ (cf. Appendix C.3). Then, by using the Lipschitz continuity of the perturbed decoder (cf. Theorem 6), we will be able to estimate the covering numbers of the adversarial hypothesis class $\widetilde{\mathcal{H}}^L$ (8) by means of the Lipschitz constants and $\mathcal{F}_\beta$.

**Lemma B.5** (Covering numbers – [23, Lemma 3.12]). *For $0 < a < \infty$, the covering numbers of the ball $B_{\|\cdot\|_{2\to2}}^{N\times n}(a) = \{X \in \mathbb{R}^{N\times n} : \|X\|_{2\to2} \le a\}$ satisfy the following for any $t > 0$:*

$$\mathcal{N}(B_{\|\cdot\|_{2\to2}}^{N\times n}(a), \|\cdot\|_{2\to2}, t) \le \left( 1 + \frac{2a}{t} \right)^{Nn}.$$

Now, we formally state Dudley's integral inequality, which we employ later on in Appendix C.4 to upper-bound the ARC.

**Theorem B.6.** *Let $(X_t)_{t\in T}$ be a random Gaussian process on a metric space $(T, d)$ with sub-gaussian increments. Then,*

$$\mathbb{E} \sup_{t\in T} X_t \le 4\sqrt{2} \int_0^{\Delta(T)/2} \sqrt{\log(\mathcal{N}(T, d, t))} dt, \tag{28}$$

*where $\Delta(T) = \sup_{t\in T} \sqrt{\mathbb{E}|X_t|^2}$.*

The next Theorem allows us to connect the ARC and the adversarial generalization error (5). Although this Theorem has been proven in the standard – non-adversarial – learning regime, we can still deploy it for our framework, since by definition of $\widetilde{\mathcal{H}}^L$ and due to the lack of the max operation in ARC, Theorem B.7 constitutes a precise tool for delivering adversarial generalization error bounds for ADMM-DAD.

**Theorem B.7** (Generalization error bounds [38, Theorem 26.5]). *Let $\mathcal{H}$ be a family of functions, $\mathcal{S}$ the training set drawn from $\mathcal{D}^s$, and $\ell$ a real-valued bounded loss function satisfying $|\ell(h, z)| \le c$, for all $h \in \mathcal{H}, z \in Z$. Then, for $\tau \in (0, 1)$, with probability at least $1 - \tau$, we have for all $h \in \mathcal{H}$*

$$\mathcal{L}_{\mathrm{true}}(h) \le \mathcal{L}_{\mathrm{train}}(h) + 2\mathcal{R}_{\mathcal{S}}(\ell \circ \mathcal{H}) + 4c\sqrt{\frac{2\log(4\tau)}{s}}, \tag{29}$$

*where*

$$\mathcal{R}_{\mathcal{S}}(\ell \circ \mathcal{H}^L) = \mathbb{E} \sup_{h\in\mathcal{H}^L} \frac{1}{s} \sum_{i=1}^{s} \epsilon_i \ell(h(y_i), x_i),$$

*is the Rademacher complexity of the hypothesis class when composed with the loss function, and $\epsilon$ is a Rademacher vector, that is, a vector with i.i.d. entries taking the values $\pm1$ with equal probability.*

To remove the dependency on the loss function $\ell(\cdot)$ and work solely with the Rademacher complexity of the hypotehesis class, we employ the well-known *contraction principle for vector-valued functions*:

**Lemma B.8** ([28, Corollary 4]). *Let $\mathcal{H}$ be a set of functions $h : \mathcal{X} \mapsto \mathbb{R}^n$, $f : \mathbb{R}^n \mapsto \mathbb{R}^n$ a K-Lipschitz function and $\mathcal{S} = \{x_i\}_{i=1}^s$. Then*

$$\mathbb{E} \sup_{h \in \mathcal{H}} \sum_{i=1}^s \epsilon_i f \circ h(x_i) \leq \sqrt{2} K \mathbb{E} \sup_{h \in \mathcal{H}} \sum_{i=1}^s \sum_{k=1}^n \epsilon_{ik} h_k(x_i), \tag{30}$$

*where $(\epsilon_i)$ and $(\epsilon_{ik})$ are Rademacher sequences.*

# C   Proofs of Main Results – Sec. 4

We dedicate this Section to the proofs of all theoretical results presented in Sec. 4.

## C.1   Proof of Proposition 5

*Proof.* We prove (12) via induction. Firstly, we notice that $\|I' + I''\|_{2\to 2} \leq \|I'\|_{2\to 2} + \|I''\|_{2\to 2} \leq 1 + \sqrt{2} = \nu$. Then, for $k = 1$:

$$\|f_W^1(Y + \Delta)\|_F \leq \nu\|B\|_F \leq \nu\|A\|_{2\to 2}\|Y + \Delta\|_F \sqrt{\beta}\|(A^T A + \rho W^T W)^{-1}\|_{2\to 2}, \tag{31}$$

which holds by definition of (1). The invertibility of $S = W^T W$ and Theorem B.1, imply that

$$\|(A^T A + \rho W^T W)^{-1}\|_{2\to 2} = \|(A^T A + \rho S)^{-1}\|_{2\to 2} \leq \frac{\rho\|S^{-1}\|_{2\to 2}}{1 - \rho\|S^{-1}\|_{2\to 2}\|A^T A\|_{2\to 2}}$$

$$= \frac{\rho}{\alpha - \rho\|A^T A\|_{2\to 2}} := \gamma, \tag{32}$$

where in the last inequality we used the fact that $\beta^{-1} \leq \|S^{-1}\|_{2\to 2} \leq \alpha^{-1}$, due to the structure of $W$ [23]. Substituting (32) into (31) yields $\|f_W^1(Y)\|_F \leq (\|Y\|_F + \|\Delta\|_F)\|A\|_{2\to 2}\nu\gamma\sqrt{\beta}$. Suppose now that (12) holds for some $k \in \mathbb{N}$. Then, for $k + 1$:

$$\begin{aligned}
\|f_W^{k+1}(Y + \Delta)\|_F &\leq \nu(\|\Theta\|_{2\to 2}\|f_W^k(Y + \Delta)\|_F + \|B\|_F) \\
&\leq \nu\left((1 + 2\|M\|_{2\to 2})\|f_W^k(Y + \Delta)\|_F + \|B\|_F\right) \\
&\leq \nu\left((1 + 2\beta\gamma\rho)\left(\nu\|A\|_{2\to 2}(\|Y + \Delta\|_F)\gamma\sqrt{\beta}\sum_{i=0}^{k-1}\nu^i(1 + 2\beta\gamma\rho)^i\right) + \|A\|_{2\to 2}\|Y + \Delta\|_F\gamma\sqrt{\beta}\right) \\
&= \|Y + \Delta\|_F\|A\|_{2\to 2}\nu\gamma\sqrt{\beta}\sum_{i=0}^k\nu^i(1 + 2\beta\gamma\rho)^i,
\end{aligned}$$

which concludes the proof. $\square$

## C.2   Proof of Theorem 6

*Proof.* Henceforth, we write $f_1^k(\cdot)$, $\Theta_1$, $M_1$, $B_1$ to denote the dependence on $W_1$ (similarly for $W_2$). Firstly, we prove Lipschitz continuity of the perturbed intermediate decoder defined in (3). Due to the explicit form of the matrices $\Theta$, $M$, $B$, the 1-Lipschitzness of $\mathcal{S}_{\lambda/\rho}(\cdot)$, Proposition 5, and the introduction of mixed terms, we obtain

$$\begin{aligned}
&\|f_1^k(Y + \Delta) - f_2^k(Y + \Delta)\|_F \\
&\quad \leq \|I'(\Theta_2 f_2^{k-1}(Y + \Delta) + B_2) + I''\mathcal{S}_{\lambda/\rho}(\Theta_2 f_2^{k-1}(Y + \Delta) + B_2)
\end{aligned}$$

$$- I'(\Theta_1 f_1^{k-1}(Y + \Delta) + B_1) - I''\mathcal{S}_{\lambda/\rho}(\Theta_1 f_1^{k-1}(Y + \Delta) + B_1)\|_F$$

$$=\|I'(\Theta_2 f_2^{k-1}(Y + \Delta) - \Theta_2 f_1^{k-1}(Y + \Delta)) + I'B_2 + I''\mathcal{S}_{\lambda/\rho}(\Theta_2 f_2^{k-1}(Y + \Delta) + B_2)$$

$$- I'(\Theta_1 f_1^{k-1}(Y + \Delta) - \Theta_2 f_1^{k-1}(Y + \Delta)) - I'B_1 - I''\mathcal{S}_{\lambda/\rho}(\Theta_1 f_1^{k-1}(Y + \Delta) + B_1)\|_F$$

$$\leq\|\Theta_2 - \Theta_1\|_{2\to2}\|f_1^{k-1}(Y + \Delta)\|_F + \|\Theta_2\|_{2\to2}\|f_2^{k-1}(Y + \Delta) - f_1^{k-1}(Y + \Delta)\|_F + \|B_2 - B_1\|_F$$

$$+ \sqrt{2}\|\Theta_2 f_2^{k-1}(Y + \Delta) + B_2 - \Theta_1 f_1^{k-1}(Y + \Delta) - B_1\|_F$$

$$\leq\|\Theta_2 - \Theta_1\|_{2\to2}\|f_1^{k-1}(Y + \Delta)\|_F + \|\Theta_2\|_{2\to2}\|f_2^{k-1}(Y + \Delta) - f_1^{k-1}(Y + \Delta)\|_F + \|B_2 - B_1\|_F$$

$$+ \sqrt{2}(\|B_2 - B_1\|_F$$

$$+ \|\Theta_2 f_2^{k-1}(Y + \Delta) - \Theta_2 f_1^{k-1}(Y + \Delta) + \Theta_2 f_1^{k-1}(Y + \Delta) - \Theta_1 f_1^{k-1}(Y + \Delta)\|_F)$$

$$\leq\|\Theta_2 - \Theta_1\|_{2\to2}\|f_1^{k-1}(Y + \Delta)\|_F + \|\Theta_2\|_{2\to2}\|f_2^{k-1}(Y + \Delta) - f_1^{k-1}(Y + \Delta)\|_F$$

$$+ \sqrt{2}(\|\Theta_2\|_{2\to2}\|f_2^{k-1}(Y + \Delta) - f_1^{k-1}(Y + \Delta)\|_F + \|\Theta_2 - \Theta_1\|_{2\to2}\|f_1^{k-1}(Y + \Delta)\|_F)$$

$$+ (1 + \sqrt{2})\|B_2 - B_1\|_F$$

$$\leq\nu\Big(\|\Theta_2 - \Theta_1\|_{2\to2}\|f_1^{k-1}(Y + \Delta)\|_F + \|\Theta_2\|_{2\to2}\|f_2^{k-1}(Y + \Delta) - f_1^{k-1}(Y + \Delta)\|_F + \|B_2 - B_1\|_F\Big) \implies$$

$$\|f_1^k(Y + \Delta) - f_2^k(Y + \Delta)\|_F$$

$$\leq\nu\Big(2\underbrace{\|M_2 - M_1\|_{2\to2}}_{(**)}\|f_1^{k-1}(Y + \Delta)\|_F \tag{33}$$

$$+ (1 + 2\beta\gamma\rho)\|f_2^{k-1}(Y + \Delta) - f_1^{k-1}(Y + \Delta)\|_F + \underbrace{\|B_2 - B_1\|_F}_{(\heartsuit)}\Big),$$

with $\nu = 1 + \sqrt{2}$ implied as in Appendix C.1.
Since the proof becomes rather technical, for the sake of readability, we separate it into corresponding subsections from that point on.

### C.2.1 Upper-bounding $(**)$

$$\|M_2 - M_1\|_{2\to2} \leq \|\rho W_2(A^TA + \rho W_2^T W_2)^{-1}W_2^T - \rho W_1(A^TA + \rho W_1^T W_1)^{-1}W_1^T\|_{2\to2}$$

$$= \rho\|W_2(A^TA + \rho W_2^T W_2)^{-1}W_2^T - W_2(A^TA + \rho W_1^T W_1)^{-1}W_2^T$$

$$+ W_2(A^TA + \rho W_1^T W_1)^{-1}W_2^T - W_1(A^TA + \rho W_1^T W_1)^{-1}W_1^T\|_{2\to2}$$

$$\leq \rho\underbrace{\|W_2[(A^TA + \rho W_2^T W_2)^{-1} - (A^TA + \rho W_1^T W_1)^{-1}]W_2^T\|_{2\to2}}_{(\dagger)}$$

$$+\rho\underbrace{\|W_2(A^TA + \rho W_1^T W_1)^{-1}W_2^T - W_1(A^TA + \rho W_1^T W_1)^{-1}W_1^T\|_{2\to2}}_{(\dagger\dagger)}.$$

According to Appendix C.1, we have

$$\|(A^TA + \rho W_1^T W_1)^{-1}\|_{2\to2} = \|(A^TA + \rho W_2^T W_2)^{-1}\|_{2\to2} = \frac{\rho}{\alpha - \rho\|A^TA\|_{2\to2}} := \gamma.$$

Therefore, for $(\dagger\dagger)$, we introduce mixed terms to obtain

$$\|W_2(A^TA + \rho W_1^T W_1)^{-1}W_2^T - W_1(A^TA + \rho W_1^T W_1)^{-1}W_1^T\|_{2\to2}$$

$$=\|W_2(A^TA + \rho W_1^T W_1)^{-1}W_2^T - W_2(A^TA + \rho W_1^T W_1)^{-1}W_1^T$$

$$+ W_2(A^TA + \rho W_1^T W_1)^{-1}W_1^T - W_1(A^TA + \rho W_1^T W_1)^{-1}W_1^T\|_{2\to2}$$

$$\leq\|W_2\|_{2\to2}\|(A^TA + \rho W_1^T W_1)^{-1}\|_{2\to2}\|W_2 - W_1\|_{2\to2}$$

$$+ \|W_1\|_{2\to2}\|(A^TA + \rho W_1^T W_1)^{-1}\|_{2\to2}\|W_2 - W_1\|_{2\to2}$$

$$\leq 2\gamma\sqrt{\beta}\|W_2 - W_1\|_{2\to 2}.$$

For the term ($\dagger$), due to Theorem B.2, we get

$$\|W_2((A^TA + \rho W_2^T W_2)^{-1} - (A^TA + \rho W_1^T W_1)^{-1})W_2^T\|_{2\to 2}$$
$$\leq \beta\|(A^TA + \rho W_2^T W_2)^{-1} - (A^TA + \rho W_1^T W_1)^{-1}\|_{2\to 2}$$
$$\leq \beta\rho\|(A^TA + \rho W_1^T W_1)^{-1}\|_{2\to 2}\|(A^TA + \rho W_2^T W_2)^{-1}\|_{2\to 2}\|W_2^T W_2 - W_1^T W_1\|_{2\to 2}$$
$$\leq 2\beta^{3/2}\gamma^2\rho\|W_2 - W_1\|_{2\to 2},$$

where in the last inequality we used the following derivation:

$$\|W_2^T W_2 - W_1^T W_1\|_{2\to 2} \leq \|W_2^T W_2 - W_2^T W_1 + W_2^T W_1 - W_1^T W_1\|_{2\to 2} \leq 2\sqrt{\beta}\|W_2 - W_1\|_{2\to 2}.$$

Overall, for ($**$), it holds:

$$\|M_2 - M_1\|_{2\to 2} \leq 2\gamma\rho\sqrt{\beta}(1 + 2\beta\gamma\rho)\|W_2 - W_1\|_{2\to 2}. \tag{34}$$

### C.2.2 Upper-bounding ($\heartsuit$)

The introduction of mixed terms and Theorem B.1 yield

$$\|B_2 - B_1\|_F = \|W_2(A^TA + \rho W_2^T W_2)^{-1}A^T(Y + \Delta_2) - W_1(A^TA + \rho W_1^T W_1)^{-1}A^T(Y + \Delta_1)\|_{2\to 2}$$
$$\leq \|A\|_{2\to 2}\|Y\|_F\|W_2(A^TA + \rho W_2^T W_2)^{-1} - W_1(A^TA + \rho W_1^T W_1)^{-1}\|_{2\to 2}$$
$$\quad + \|A\|_{2\to 2}\|W_2(A^TA + \rho W_2^T W_2)^{-1}\Delta_2 - W_1(A^TA + \rho W_1^T W_1)^{-1}\Delta_1\|_{2\to 2}$$
$$\leq \|A\|_{2\to 2}\|Y\|_F\|W_2(A^TA + \rho W_2^T W_2)^{-1} - W_2(A^TA + \rho W_1^T W_1)^{-1}$$
$$\quad + W_2(A^TA + \rho W_1^T W_1)^{-1} - W_1(A^TA + \rho W_1^T W_1)^{-1}\|_{2\to 2}$$
$$\quad + \|A\|_{2\to 2}\|W_2(A^TA + \rho W_2^T W_2)^{-1}\Delta_2 - W_2(A^TA + \rho W_2^T W_2)^{-1}\Delta_1$$
$$\quad + W_2(A^TA + \rho W_2^T W_2)^{-1}\Delta_1 - W_1(A^TA + \rho W_1^T W_1)^{-1}\Delta_1\|_{2\to 2}$$
$$\leq \|A\|_{2\to 2}\Big(\|Y\|_F\Big(\sqrt{\beta}\|(A^TA + \rho W_2^T W_2)^{-1} - (A^TA + \rho W_1^T W_1)^{-1}\|_{2\to 2} + \gamma\|W_2 - W_1\|_{2\to 2}\Big)$$
$$\quad + \sqrt{\beta}\gamma\|\Delta_2 - \Delta_1\|_F + \|\Delta_1\|_F\|W_2(A^TA + \rho W_2^T W_2)^{-1} - W_1(A^TA + \rho W_1^T W_1)^{-1}\|_{2\to 2}\Big)$$
$$\leq \|A\|_{2\to 2}\Big(\|Y\|_F\Big(\sqrt{\beta}\rho\|(A^TA + \rho W_1^T W_1)^{-1}\|_{2\to 2}\|(A^TA + \rho W_2^T W_2)^{-1}\|_{2\to 2}\|W_2^T W_2 - W_1^T W_1\|_{2\to 2}$$
$$\quad + \gamma\|W_2 - W_1\|_{2\to 2}\Big) + \sqrt{\beta}\gamma\|\Delta_2 - \Delta_1\|_F$$
$$\quad + \|\Delta_1\|_F\|(A^TA + \rho W_2^T W_2)^{-1} - (A^TA + \rho W_1^T W_1)^{-1}\|_{2\to 2}\Big)$$
$$\leq \|A\|_{2\to 2}\Big(\|Y\|_F\Big(\sqrt{\beta}\rho\|(A^TA + \rho W_1^T W_1)^{-1}\|_{2\to 2}\|(A^TA + \rho W_2^T W_2)^{-1}\|_{2\to 2}\|W_2^T W_2 - W_1^T W_1\|_{2\to 2}$$
$$\quad + \gamma\|W_2 - W_1\|_{2\to 2}\Big) + \sqrt{\beta}\gamma\|\Delta_2 - \Delta_1\|_F$$
$$\quad + \rho\|\Delta_1\|_F\|(A^TA + \rho W_1^T W_1)^{-1}\|_{2\to 2}\|(A^TA + \rho W_2^T W_2)^{-1}\|_{2\to 2}\|W_2^T W_2 - W_1^T W_1\|_{2\to 2}\Big)$$

All in all, for $\|B_2 - B_1\|_F$, we obtain:

$$
\begin{aligned}
\|B_2 - B_1\|_F \leq &\gamma\|A\|_{2\to2}\Big(\|Y\|_F(1 + 2\beta\gamma\rho)\|W_2 - W_1\|_{2\to2} \\
&+ \sqrt{\beta}\underbrace{\|\Delta_2 - \Delta_1\|_F}_{(\diamond)} + E(1 + 2\beta\gamma\rho)\|W_2 - W_1\|_{2\to2}\Big),
\end{aligned}
\tag{35}
$$

where $E := \sqrt{s}\varepsilon$.

### C.2.3  Upper-bounding $(\diamond)$

We write $\ell(\cdot) = \|\cdot\|_2^2$, to make notation more compact for the time being. By definition of $\Delta_i$, $i = 1, 2$, we have

$$
E\left\|\frac{\nabla_Y\ell(f_2^k(Y), X)}{\|\nabla_Y\ell(f_2^k(Y), X)\|_F} - \frac{\nabla_Y\ell(f_1^k(Y), X)}{\|\nabla_Y\ell(f_1^k(Y), X)\|_F}\right\|_F.
\tag{36}
$$

Consequently, for (36), due to **Assumptions (a)** and **(c)** of Sec. 3, we get

$$
\begin{aligned}
&E\left\|\frac{\nabla_Y\ell(f_2^k(Y), X)}{\|\nabla_Y\ell(f_2^k(Y), X)\|_F} - \frac{\nabla_Y\ell(f_1^k(Y), X)}{\|\nabla_Y\ell(f_1^k(Y), X)\|_F}\right\|_F \\
&\leq E\left\|\frac{\|\nabla_Y\ell(f_1^k(Y), X)\|_F\nabla_Y\ell(f_2^k(Y), X) - \|\nabla_Y\ell(f_2^k(Y), X)\|_F\nabla_Y\ell(f_1^k(Y), X)}{\|\nabla_Y\ell(f_2^k(Y), X)\|_F\|\nabla_Y\ell(f_1^k(Y), X)\|_F}\right\|_F \\
&\leq \frac{E}{\kappa^2}\|\ \|\nabla_Y\ell(f_1^k(Y), X)\|_F\nabla_Y\ell(f_2^k(Y), X) - \|\nabla_Y\ell(f_1^k(Y), X)\|_F\nabla_Y\ell(f_1^k(Y), X) \\
&\quad + \|\nabla_Y\ell(f_1^k(Y), X)\|_F\nabla_Y\ell(f_1^k(Y), X) - \|\nabla_Y\ell(f_2^k(Y), X)\|_F\nabla_Y\ell(f_1^k(Y), X)\ \|_F \\
&\leq \frac{E}{\kappa^2}\Big(\|\nabla_Y\ell(f_1^k(Y), X)\|_F \cdot \|\nabla_Y\ell(f_2^k(Y), X) - \nabla_Y\ell(f_1^k(Y), X)\|_F \\
&\quad + \|\nabla_Y\ell(f_1^k(Y), X)\| \cdot \|\ \|\nabla_Y\ell(f_2^k(Y), X)\|_F - \|\nabla_Y\ell(f_1^k(Y), X)\|_F\ \|_F\Big) \\
&\leq \frac{2E\|\nabla_Y\ell(f_1^k(Y), X)\|_F}{\kappa^2}\|\nabla_Y\ell(f_2^k(Y), X) - \nabla_Y\ell(f_1^k(Y), X)\|_F \\
&\leq \frac{2E(B_{\text{in}} + B_{\text{out}})\|\nabla_Y f_1^k(Y)\|_2}{\kappa^2}\underbrace{\|\nabla_Y\ell(f_2^k(Y), X) - \nabla_Y\ell(f_1^k(Y), X)\|_F}_{(\text{T.1})},
\end{aligned}
\tag{37}
$$

where in the last inequality we used the derivation $\nabla_Y\ell(f_W^k(Y), X) = 2(f_W^k(Y) - X)\nabla_Y(f_W^k(Y))^T$, for all $W \in \mathcal{F}_\beta$. Now, for (**T.1**), we get

$$
\begin{aligned}
&\|2(f_2^k(Y) - X)\nabla_Y(f_2^k(Y))^T - 2(f_1^k(Y) - X)\nabla_Y(f_1^k(Y))^T\|_F \\
&\quad \leq \|2(f_2^k(Y) - X)\nabla_Y(f_2^k(Y))^T - 2(f_1^k(Y) - X)\nabla_Y(f_2^k(Y))^T \\
&\qquad + 2(f_1^k(Y) - X)\nabla_Y(f_2^k(Y))^T - 2(f_1^k(Y) - X)\nabla_Y(f_1^k(Y))^T\|\|_F \\
&\quad \leq 2(B_{\text{in}} + B_{\text{out}})\|\nabla_Y(f_2^k(Y))^T - \nabla_Y(f_1^k(Y))^T\|_F + 2\|\nabla_Y(f_2^k(Y))^T\|\|f_2^k(Y) - f_1^k(Y)\|_F \\
&\quad \leq 2\|A\|_{2\to2}\nu\gamma\sqrt{\beta}\Sigma_k\|W_2 - W_1\|_{2\to2}\sum_{i=0}^{k-2}\nu^i(1 + 2\beta\gamma\rho)^i \\
&\qquad + 2(B_{\text{in}} + B_{\text{out}})\underbrace{\|\nabla_Y(f_2^k(Y) - f_1^k(Y))\|_F}_{(\text{T.2})},
\end{aligned}
\tag{38}
$$

where in the last inequality we used Proposition B.3 and Theorem B.4 – the Lipschitz continuity of $f_W^k(Y)$ with respect to $W$, with $\Sigma_k$ being the Lipschitz constants up to an arbitrary layer $k$.

### C.2.4 Upper-bounding (T.2)

According to **Assumption (*b*)** of Sec. 3, and due to the chain rule for composite functions, for the gradient of the soft-thresholding operator with respect to $Y$ calculated at $\theta := \Theta f_W^{k-1}(Y) + B(Y)$, for any $W \in \mathcal{F}_\beta$, we have:

$$\nabla_Y(\mathcal{S}_{\lambda/\rho}(\theta)) = \begin{cases} \Theta \nabla_Y f_W^{k-1}(Y) + \nabla_Y B(Y), & \theta > \lambda/\rho \\ 0, & \theta \le \lambda/\rho. \end{cases} \tag{39}$$

We calculate $\nabla_Y B(Y) = W(A^T A + \rho W^T W)^{-1} A^T$, and assume without loss of generality that the first clause of (39) holds, since otherwise we simply get rid of an extra $\sqrt{2}$ term. The, the introduction of mixed terms, and the application of Theorem B.1 and Proposition B.3 yield

$$\|\nabla_Y f_2^k(Y) - \nabla_Y f_1^k(Y)\|_2$$

$$= \|\nabla_Y \Big( I'(\Theta_2 f_2^{k-1}(Y) + B_2(Y)) + I'' \mathcal{S}_{\lambda/\rho}(\Theta_2 f_2^{k-1}(Y) + B_2(Y)) \Big)$$

$$\quad - \nabla_Y \Big( I'(\Theta_1 f_1^{k-1}(Y) + B_1(Y)) + I'' \mathcal{S}_{\lambda/\rho}(\Theta_1 f_1^{k-1}(Y) + B_1(Y)) \Big) \|_F$$

$$= \| I' \Theta_2 \nabla_Y f_2^{k-1}(Y) + I' \nabla_Y B_2(Y) + I''(\Theta_2 \nabla_Y f_2^{k-1}(Y) + \nabla_Y B_2(Y))$$

$$\quad - I' \Theta_1 \nabla_Y f_1^{k-1}(Y) - I' \nabla_Y B_1(Y) - I''(\Theta_1 \nabla_Y f_1^{k-1}(Y) + \nabla_Y B_1(Y)) \|_F$$

$$= \| \Theta_2(I' + I'')\nabla_Y f_2^{k-1}(Y) - \Theta_1(I' + I'')\nabla_Y f_1^{k-1}(Y) + (I' + I'')\nabla_Y B_2(Y) - (I' + I'')\nabla_Y B_1(Y)\|_F$$

$$\le \| \Theta_2(I' + I'')\nabla_Y f_2^{k-1}(Y) - \Theta_2(I' + I'')\nabla_Y f_1^{k-1}(Y)$$

$$\quad + \Theta_2(I' + I'')\nabla_Y f_1^{k-1}(Y) - \Theta_1(I' + I'')\nabla_Y f_1^{k-1}(Y)\|_F + \|(I' + I'')(\nabla_Y B_2(Y) - \nabla_Y B_1(Y))\|_F$$

$$\le \nu \|\Theta_2\|_{2\to2}\|\nabla_Y f_2^{k-1}(Y) - \nabla_Y f_1^{k-1}(Y)\|_F + 2\nu\|\nabla_Y f_1^{k-1}(Y)\|_F\|M_2 - M_1\|_{2\to2} + \nu\|\nabla_Y B_2(Y) - \nabla_Y B_1(Y)\|_F$$

$$\overset{(34)}{\le} \nu(1 + 2\beta\gamma\rho)\|\nabla_Y f_2^{k-1}(Y) - \nabla_Y f_1^{k-1}(Y)\|_F + 4\nu\gamma\rho\sqrt{\beta}(1 + 2\beta\gamma\rho)\|\nabla_Y f_1^{k-1}(Y)\|_F\|W_2 - W_1\|_{2\to2}$$

$$\quad + \nu\|A\|_{2\to2}\|W_2(A^T A + \rho W_2^T W_2) - W_1(A^T A + \rho W_1^T W_1)\|_{2\to2}$$

$$\le \nu(1 + 2\beta\gamma\rho)\|\nabla_Y f_2^{k-1}(Y) - \nabla_Y f_1^{k-1}(Y)\|_F + 4\nu\gamma\rho\sqrt{\beta}(1 + 2\beta\gamma\rho)\|\nabla_Y f_1^{k-1}(Y)\|_F\|W_2 - W_1\|_{2\to2}$$

$$\quad + \nu\|A\|_{2\to2}\|W_2(A^T A + \rho W_2^T W_2) - W_1(A^T A + \rho W_2^T W_2)$$

$$\quad + W_1(A^T A + \rho W_2^T W_2) - W_1(A^T A + \rho W_1^T W_1)\|_{2\to2}$$

$$\le \nu(1 + 2\beta\gamma\rho)\|\nabla_Y f_2^{k-1}(Y) - \nabla_Y f_1^{k-1}(Y)\|_F + 4\nu\gamma\rho\sqrt{\beta}(1 + 2\beta\gamma\rho)\|\nabla_Y f_1^{k-1}(Y)\|_F\|W_2 - W_1\|_{2\to2}$$

$$\quad + \nu\|A\|_{2\to2}\Big(\gamma\|W_2 - W_1\|_{2\to2} + \sqrt{\beta}\rho\gamma^2\|W_2 - W_1\|_{2\to2}\Big)$$

$$\le \nu(1 + 2\beta\gamma\rho)\|\nabla_Y f_2^{k-1}(Y) - \nabla_Y f_1^{k-1}(Y)\|_F$$

$$\quad + 8\nu^2\|A\|_{2\to2}\gamma\rho\sqrt{\beta}(\gamma\sqrt{\beta} + \beta^{3/2}\gamma^2\rho)\|W_2 - W_1\|_{2\to2}\sum_{i=0}^{k-2} \nu^i(1 + 2\beta\gamma\rho)^i$$

$$\quad + \nu\gamma\|A\|_{2\to2}(1 + 2\beta\gamma\rho)\|W_2 - W_1\|_{2\to2}$$

$$\le r\nu\Big(\|\nabla_Y f_2^{k-1}(Y) - \nabla_Y f_1^{k-1}(Y)\|_F + 8\nu\gamma^2\rho\beta\|A\|_{2\to2}\|W_2 - W_1\|_{2\to2}\sum_{i=0}^{k-2}(r\nu)^i$$

$$\quad + \gamma\|A\|_{2\to2}\|W_2 - W_1\|_{2\to2}\Big),$$

where we set $r = 1 + 2\beta\gamma\rho$. Now, for all $k \ge 1$, we define

$$G = rv, \tag{40}$$

$$D_k = \sum_{i=0}^{k-1} G^i, \quad D_0 = 0, \tag{41}$$

$$Z_k = G(8v\gamma^2\rho\beta\|A\|_{2\to2})D_{k-1} + \gamma\|A\|_{2\to2}, \tag{42}$$

so that

$$\|\nabla_Y f_2^k(Y) - \nabla_Y f_1^k(Y)\|_2 \le G\|\nabla_Y f_2^{k-1}(Y) - \nabla_Y f_1^{k-1}(Y)\|_F + Z_k\|W_2 - W_1\|_{2\to2}. \tag{43}$$

We prove via induction that

$$C_L = \sum_{k=1}^{L} G^{L-k}Z_k, \quad L \ge 1. \tag{44}$$

First, notice that for $L = 1$, it holds

$$\begin{aligned}
\|\nabla_Y f_{W_2}^1(Y) - \nabla_Y f_{W_1}^1(Y)\|_F &= \|I'\nabla_Y B_1(Y) + I''\nabla_Y(\mathcal{S}_{\lambda/\rho}(B_1)) - I'\nabla_Y B_2 - I''\nabla_Y(\mathcal{S}_{\lambda/\rho}(B_2))\|_F \\
&\le v\|B_2 - B_1\|_F \\
&\le rv\gamma\|A\|_{2\to2}\|W_2 - W_1\|_{2\to2} \\
&= Z_1\|W_2 - W_1\|_{2\to2} \\
&= C_1\|W_2 - W_1\|_{2\to2},
\end{aligned}$$

so $C_1$ has indeed the form described in (44). Suppose that (44) holds for some $L \in \mathbb{N}$. Then, for $L + 1$:

$$\begin{aligned}
\|\nabla_Y f_2^{L+1}(Y) - \nabla_Y f_1^{L+1}(Y)\|_F &\le G\|\nabla_Y f_2^L(Y) - \nabla_Y f_1^L(Y)\|_F + Z_{L+1}\|W_2 - W_1\|_{2\to2} \\
&\le (GC_L + Z_{L+1})\|W_2 - W_1\|_{2\to2} \\
&= \left(G\sum_{k=1}^{L} G^{L-k}Z_k + Z_{L+1}\right)\|W_2 - W_1\|_{2\to2} \\
&= \left(\sum_{k=1}^{L+1} G^{L-k}Z_k\right)\|W_2 - W_1\|_{2\to2} \\
&= C_{L+1}\|W_2 - W_1\|_{2\to2},
\end{aligned}$$

which proves that for any $L \in \mathbb{N}$, it holds

$$\|\nabla_Y f_{W_2}^L(Y) - \nabla_Y f_{W_1}^L(Y)\|_F \le C_L\|W_2 - W_1\|_{2\to2}. \tag{45}$$

We combine the results from Sec. C.2.3 – C.2.4 to deduce:

$$\begin{aligned}
\|\Delta_2 - \Delta_1\|_F \le{}& 2\left(\frac{E(\mathrm{B}_{\mathrm{in}} + \mathrm{B}_{\mathrm{out}})}{\kappa^2}\|A\|_{2\to2}v\gamma\sqrt{\beta}D_k\right)\left(\|A\|_{2\to2}v\gamma\sqrt{\beta}\Sigma_k D_{k-1}\right. \\
&\left. + (\mathrm{B}_{\mathrm{in}} + \mathrm{B}_{\mathrm{out}})C_k\right)\|W_2 - W_1\|_{2\to2}.
\end{aligned} \tag{46}$$

Hence, applying (46) in (35) of Sec. C.2.2 yields

$$\|B_2 - B_1\|_F \leq \gamma \|A\|_{2\to 2} \left[ r\|Y\|_F + rE + 2\sqrt{\beta} \left( \frac{E(B_{\text{in}} + B_{\text{out}})}{\kappa^2} \|A\|_{2\to 2} \nu \gamma \sqrt{\beta} D_k \right) \right. \tag{47}$$
$$\left. \cdot \left( \|A\|_{2\to 2} \nu \gamma \sqrt{\beta} \Sigma_k D_{k-1} + (B_{\text{in}} + B_{\text{out}}) C_k \right) \right] \|W_2 - W_1\|_{2\to 2}.$$

Now, we plug (34) and (47) in (33) at the beginning of this proof to obtain

$$\|f_1^k(Y + \Delta_1) - f_2^k(Y + \Delta_2)\|_F \leq r\nu \|f_2^{k-1}(Y + \Delta) - f_1^{k-1}(Y + \Delta)\|_F + \|A\|_{2\to 2} \Big( 4r\nu^2 \gamma^2 \rho \beta D_{k-1}$$
$$+ \gamma \left[ r\|Y\|_F + rE + 2\sqrt{\beta} \left( \frac{E(B_{\text{in}} + B_{\text{out}})}{\kappa^2} \|A\|_{2\to 2} \nu \gamma \sqrt{\beta} D_k \right) \right. \tag{48}$$
$$\left. \cdot \left( \|A\|_{2\to 2} \nu \gamma \sqrt{\beta} \Sigma_k D_{k-1} + (B_{\text{in}} + B_{\text{out}}) C_k \right) \right] \Big) \|W_2 - W_1\|_{2\to 2}.$$

In order to treat all layers in a uniform manner, we set $f_1^0(Y + \Delta) = f_2^0(Y + \Delta) = Y + \Delta$. Similarly to our derivation for $C_L$ (44), we set

$$H_k = \gamma \|A\|_{2\to 2} \Big( 4r\nu^2 \beta \gamma \rho D_{k-1} + \left[ r\|Y\|_F + rE + 2\sqrt{\beta} \left( \frac{E(B_{\text{in}} + B_{\text{out}})}{\kappa^2} \|A\|_{2\to 2} \nu \gamma \sqrt{\beta} D_k \right) \right.$$
$$\left. \cdot \left( \|A\|_{2\to 2} \nu \gamma \sqrt{\beta} \Sigma_k D_{k-1} + (B_{\text{in}} + B_{\text{out}}) C_k \right) \right] \Big), \qquad k \geq 1, \tag{49}$$

with $\Sigma_k$, $D_k$, $C_k$ defined in (26), (41), (44), respectively. Now, it is a matter of calculations to prove via induction that

$$K_L' = \sum_{k=1}^{L} G^{L-k} H_k, \quad L \geq 1. \tag{50}$$

First, for $L = 1$, due to (1) and (47), we have

$$\|f_1^1(Y + \Delta_1) - f_2^1(Y + \Delta_2)\|_F$$
$$\leq \|I'B_1 + I''\mathcal{S}_{\lambda/\rho}(B_1) - I'B_2 - I''\mathcal{S}_{\lambda/\rho}(B_2)\|_F$$
$$\leq \nu \|B_2 - B_1\|_F$$
$$\leq \nu \gamma \|A\|_{2\to 2} \left[ r\|Y\|_F + rE + 2\sqrt{\beta} \left( \frac{E(B_{\text{in}} + B_{\text{out}})}{\kappa^2} \|A\|_{2\to 2} \nu \gamma \sqrt{\beta} D_k \right) \right.$$
$$\left. \cdot \left( \|A\|_{2\to 2} \nu \gamma \sqrt{\beta} \Sigma_k D_{k-1} + (B_{\text{in}} + B_{\text{out}}) C_k \right) \right] \|W_2 - W_1\|_{2\to 2}$$
$$\leq H_1 \|W_2 - W_1\|_{2\to 2}$$
$$= K_1' \|W_2 - W_1\|_{2\to 2}.$$

so $K_1'$ has indeed the form described in (50). Let us suppose that (50) holds for some $L \in \mathbb{N}$. Then, for $L + 1$:

$$\|f_2^{L+1}(Y + \Delta_2) - f_1^{L+1}(Y + \Delta_1)\|_F \leq G\|f_2^L(Y + \Delta_2) - f_1^L(Y + \Delta_1)\|_F + H_{L+1} \|W_2 - W_1\|_{2\to 2}$$
$$\leq (GK_L' + H_{L+1}) \|W_2 - W_1\|_{2\to 2}$$

$$= \left( G \sum_{k=1}^{L} G^{L-k} H_k + H_{L+1} \right) \|W_2 - W_1\|_{2\to 2}$$

$$= \left( \sum_{k=1}^{L+1} G^{L-k} H_k \right) \|W_2 - W_1\|_{2\to 2}$$

$$= K'_{L+1} \|W_2 - W_1\|_{2\to 2}.$$

Therefore, for any $L \in \mathbb{N}$, it holds

$$\|f_2^L(Y + \Delta_2) - f_1^L(Y + \Delta_1)\|_F \le K'_L \|W_2 - W_1\|_{2\to 2}, \tag{51}$$

with $K'_L$ defined as in (50) This means that the perturbed intermediate decoder (6) is Lipschitz continuous with respect to $W$. For the perturbed final decoder (7), the affine map $T_W$ is by definition Lipschitz continuous, with Lipschitz constant satisfying

$$\text{Lip}_{T_{W_1}} = \|T_{W_1}\|_{2\to 2} = \|T_{W_2}\|_{2\to 2} = \text{Lip}_{T_{W_2}} \le 2\gamma\rho\sqrt{\beta}. \tag{52}$$

Therefore, we introduce mixed terms to get:

$$\|T_{W_2}(f_{W_2}^L(Y + \Delta_2)) - T_{W_1}(f_{W_1}^L(Y + \Delta_1))\|_F$$

$$= \|T_{W_2}(f_{W_2}^L(Y + \Delta_2)) - T_{W_2}(f_{W_1}^L(Y + \Delta_1)) + T_{W_2}(f_{W_1}^L(Y + \Delta_1)) - T_{W_1}(f_{W_1}^L(Y + \Delta_1))\|_F$$

$$\le \|T_{W_2}\|_{2\to 2} \|f_{W_2}^L(Y + \Delta_2)) - f_{W_1}^L(Y + \Delta_1))\|_F + \|T_{W_2} - T_{W_1}\|_{2\to 2} \|f_{W_1}^L(Y + \Delta_1))\|_F$$

$$\le 2\gamma\rho\sqrt{\beta}K'_L \|W_2 - W_1\|_{2\to 2} + \left( \nu\|A\|_{2\to 2}(\|Y\|_F + E)\gamma\sqrt{\beta}D_L \right) \|T_{W_2} - T_{W_1}\|_{2\to 2}$$

$$\le 2\gamma\rho\sqrt{\beta}K'_L \|W_2 - W_1\|_{2\to 2}$$

$$+ 2\rho\left( \nu\|A\|_{2\to 2}(\|Y\|_F + E)\gamma\sqrt{\beta}D_L \right) \|(A^T A + \rho W_2^T W_2)^{-1} W_2^T W_2 - (A^T A + \rho W_1^T W_1)^{-1} W_1^T W_1\|_{2\to 2}$$

$$\le \underbrace{\left( 2\gamma\rho\sqrt{\beta}K'_L + 2\nu^2\gamma^2\rho\|A\|_{2\to 2}(\|Y\|_F + E)\sqrt{\beta}D_L \right)}_{:=\text{Lip}_h^{L,\varepsilon}} \|W_2 - W_1\|_{2\to 2}. \tag{53}$$

Overall, the perturbed final decoder is Lipschitz continuous, and we denote its Lipschitz constants with $\text{Lip}_h^{L,\varepsilon}$, to indicate the dependence on both $L$ and $\varepsilon$. Consequently, we have proven that, for all $L \ge 2$,

$$\|h_{W_1}^L(Y + \Delta_1) - h_{W_2}^L(Y + \Delta_2)\|_F \le \text{Lip}_h^{L,\varepsilon}\|W_1 - W_2\|_{2\to 2}, \tag{54}$$

where

$$\text{Lip}_h^{L,\varepsilon} = 2\gamma\rho\sqrt{\beta}\bigg( (r\nu)^{L-1}\gamma\|A\|_{2\to 2}\Big( r\|Y\|_F + rE + 2\beta(B_{\text{in}} + B_{\text{out}})^2 \frac{E}{\kappa^2}\nu\gamma^2\|A\|_{2\to 2}^2 \Big)$$

$$+ \sum_{k=2}^{L} (r\nu)^{L-k} H_k + \nu^2\gamma\|A\|_{2\to 2}(\|Y\|_F + E)\Big( 1 + \sum_{k=1}^{L-1}(r\nu)^k \Big) \bigg), \tag{‡}$$

with $E = \sqrt{s}\varepsilon$, $\nu = (1 + \sqrt{2})$, $r = 1 + 2\beta\gamma\rho$, $\gamma$ as in Proposition 5, $\beta$ as in Definition 1, and $H_k$ defined in (49). $\qquad\square$

### C.3 Proof of Proposition 7.

*Proof.* By Definition 1, we have $\mathcal{F}_\beta \subset B^{N\times n}_{\|\cdot\|_{2\to 2}}(\sqrt{\beta})$. Then, the application of Lemma B.5 for $\mathcal{F}_\beta$ implies that

$$\mathcal{N}(\mathcal{F}_\beta, \|\cdot\|_{2\to 2}, t) \le \left(1 + \frac{2\sqrt{\beta}}{t}\right)^{Nn}. \tag{55}$$

Therefore, due to Theorem 6, the covering numbers of $\widetilde{\mathcal{M}}$ are bounded as follows:

$$\mathcal{N}(\widetilde{\mathcal{M}}, \|\cdot\|_F, t) \le \mathcal{N}(\mathrm{Lip}^{L,\varepsilon}_h \mathcal{F}_\beta, \|\cdot\|_{2\to 2}, t) = \mathcal{N}(\mathcal{F}_\beta, \|\cdot\|_{2\to 2}, t/\mathrm{Lip}^{L,\varepsilon}_h) \le \left(1 + \frac{2\sqrt{\beta}\mathrm{Lip}^{L,\varepsilon}_h}{t}\right)^{Nn}. \tag{56}$$

□

### C.4 Proof of Theorem 8

*Proof.* The ARC has sub-gaussian increments, so we can use Dudley's integral inequality (28) to upper bound it in terms of the covering numbers of the set $\widetilde{\mathcal{M}}$ defined in Sec. 4. To that end, we first calculate

$$\Delta(\widetilde{\mathcal{M}}) = \sup_{\tilde{h}\in\widetilde{\mathcal{H}}^L} \sqrt{\mathbb{E}\left(\sum_{i=1}^s \sum_{k=1}^n \epsilon_{ik}\tilde{h}_k(y_i)\right)^2} \le \sup_{\tilde{h}\in\widetilde{\mathcal{H}}^L} \sqrt{\mathbb{E}\sum_{i=1}^s \sum_{k=1}^n \epsilon_{ik}(\tilde{h}_k(y_i))^2}$$

$$\le \sup_{\tilde{h}\in\widetilde{\mathcal{H}}^L} \sqrt{\sum_{i=1}^s \|\tilde{h}(y_i)\|_2^2} \le \sqrt{s}B_{\mathrm{out}}. \tag{57}$$

Then, we combine Proposition 7 and Theorem B.6 to get:

$$\mathcal{R}_{\mathbf{S}}(\widetilde{\mathcal{H}}^L) \le \frac{4\sqrt{2}}{s} \int_0^{\frac{\sqrt{s}B_{\mathrm{out}}}{2}} \sqrt{\log\mathcal{N}(\widetilde{\mathcal{M}}, \|\cdot\|_F, t)}dt$$

$$\le O\left(\int_0^{\frac{\sqrt{s}B_{\mathrm{out}}}{2}} \sqrt{Nn\log\left(1 + \frac{2\sqrt{\beta}\mathrm{Lip}^{L,\varepsilon}_h}{t}\right)}dt\right), \tag{58}$$

which is the desired estimate. □

### C.5 Proof of Theorem 3

*Proof.* Due to (18), and the inequality [11, Lemma C.9]

$$\int_0^a \sqrt{\log\left(1 + \frac{b}{t}\right)}dt \le a\sqrt{\log(e(1 + b/a))}, \qquad a, b > 0, \tag{59}$$

the following holds for the ARC:

$$\mathcal{R}_{\mathbf{S}}(\|\cdot\|_2^2 \circ \widetilde{\mathcal{H}}^L) \le O\left(\sqrt{\frac{Nn}{s}}\sqrt{\log\left(\exp\cdot\left(1 + \frac{2\sqrt{\beta}\mathrm{Lip}^{L,\varepsilon}_h}{\sqrt{s}B_{\mathrm{out}}}\right)\right)}\right). \tag{60}$$

According to **Assumption (a)**, we deduce that the loss function $\|\cdot\|_2^2$ is upper-bounded by $c = (B_{\mathrm{in}} + B_{\mathrm{out}})^2$. The result follows by substituting (60) in Theorem B.7, with the aforesaid $c$. □

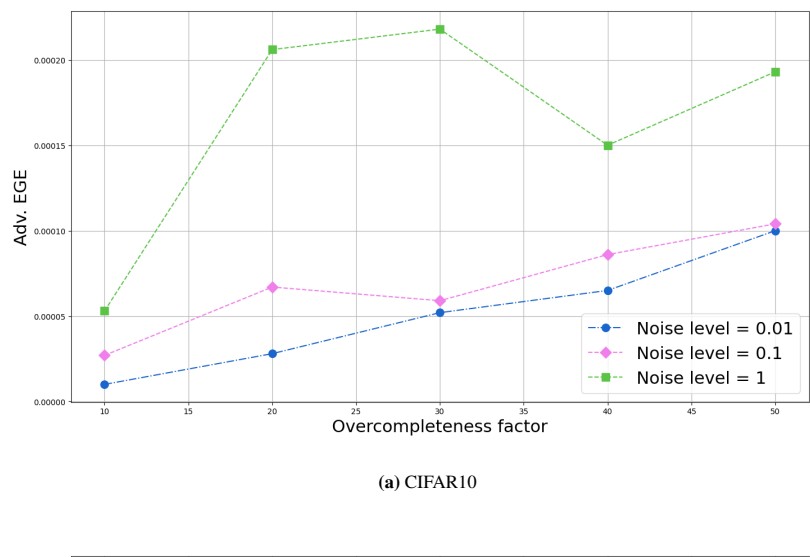

**(a)** CIFAR10

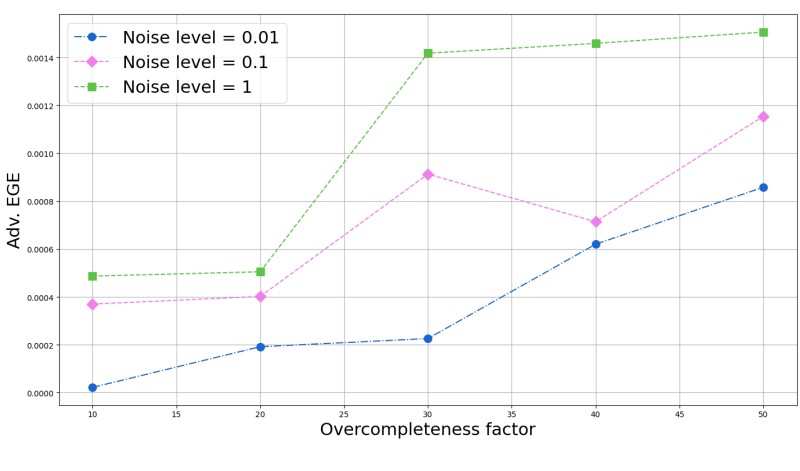

**(b)** SVHN

**Figure 4:** Adversarial generalization of ADMM-DAD measured in terms of the adversarial empirical generalization error (21), for alternating overcompleteness $N$ and different attack levels $\varepsilon$. For both datasets, (21) increases as $N$ also increases, like Theorem 3 suggests, thus confirming our derived adversarial generalization theory.

# D    Experimental details

To encourage reproducibility of our results, we hereby complement the experimental settings of Sec. 5.

We follow a standard CS setup and employ a normal Gaussian observation matrix $A \in \mathbb{R}^{m \times n}$, which we normalize as $A/\sqrt{m}$ for the CIFAR10 dataset, and as $A^T A = I_{n \times n}$ for the SVHN dataset. The experimental parameters $\lambda$ and $\rho$ have been calibrated accordingly, to account for the different structural specifications of each dataset. Therefore, for CIFAR10, we set $\rho = 1$ and $\lambda = 10^{-4}$, while for SVHN we alternate $\rho$ and $\lambda$ depending on the value of $N$. Particularly, for $N = [10, 20, 30, 40, 50]$, we set $\lambda = [10^{-5}, 10^{-4}, 10^{-4}, 10^{-3}, 10^{-5}]$ and $\rho = [100, 1, 1, 1, 10]$, respectively.

For all implementations, we employ the Adam algorithm [21], which constitutes a stochastic optimization method that adaptively estimates lower-order moments of the gradient of the adversarial training MSE. All of Adam's parameters are set to their default values, except for the learning rate $\epsilon_{\text{lr}}$. Specifically, for the CIFAR10 dataset, we train the 5- and 10-layer ADMM-DAD with $\epsilon_{\text{lr}} = 10^{-5}$ and $\epsilon_{\text{lr}} = 10^{-4}$, respectively. For the SVHN dataset, we train the 10- and 15-layer ADMM-DAD with $\epsilon_{\text{lr}} = 10^{-4}$ and $\epsilon_{\text{lr}} = 10^{-5}$, respectively. We train all models on all datasets using early-stopping with

respect to the adversarial empirical generalization error (adversarial EGE) (21). We repeat all the experiments at least 10 times and average the results over the runs. For the comparisons with the baseline ISTA-net, we set the best hyper-parameters proposed by the original authors. For the course of our experiments, we have utilized a node of 4 H100 GPUs.

**Adversarial generalization error with alternating** $N$**.** For the sake of completeness, we present in Figure 4 the scaling of the adversarial EGE (21), corresponding to the clean test MSEs (19) and the adversarial test MSEs (20) depicted in Figure 3, for increasing $N$, and three different values of $\varepsilon$. Similarly to our discussion in Sec. 5, we observe that the adversarial EGE increases as $N$ and $\varepsilon$ increase, for both datasets, thereby corroborating our theoretical derivations for the adversarial generalization of ADMM-DAD.

# E   Impact Statement

Our work contributes to the theoretical understanding of adversarial robustness in DUNs, which are designed to solve inverse problems like CS. While the research is primarily theoretical, it provides key insights that could help improve the reliability and robustness of neural networks in high-stakes applications, such as medical imaging. Given the theoretical and exploratory nature of our study, it does not pose any foreseeable societal risks in the near term. Instead, it lays the groundwork for future robust machine learning systems, enjoying enhanced interpretability and resilience to adversarial attacks.

