# OpenReview forum: "Adversarial generalization of unfolding (model-based) networks"
_NeurIPS.cc/2025/Conference — NeurIPS 2025 poster_

### Official Review · Reviewer_PrVf · 2025-06-12

**Clarity:** 3
**Significance:** 2
**Originality:** 3
**Rating:** 4
**Confidence:** 2

**Summary:**

This work presents the first adversarial generalization bound for unfolding networks—a class of neural networks with structured architectures tailored for specific inverse problems. The authors assume perturbations are bounded in the $l_2$ norm and are generated using the Fast Gradient Sign Method (FGSM). Under these assumptions, they provide the first theoretical analysis of the adversarial generalization properties of unfolding networks.

**Questions:**

Can the bound be extended to the precise attack rather than FGSM attack?

**Ethical Concerns:**

["NO or VERY MINOR ethics concerns only"]

**Limitations:**

Yes.

**Quality:**

3

**Strengths And Weaknesses:**

Strengths
1. This paper is well-written and easy to follow.
2. The proposed bound is tighter than before.

Weaknesses
1. The authors assume the FGSM attack, which is a weak attack in practice. It seems that the compared bound [1] did not assume the FGSM attack but used the precise maximal value.
2. It would be better to explain what the overcomplete sparsifier is.
3. There are some typos:

	Line 120, $O_{n\times n}\in \mathbb{R}^n$ should be $O_{n\times n}\in \mathbb{R}^{n\times n}$

	Line 169, it lacks a right bracket.

	Equation 10, why is there a $log(exp(\cdots))$, which can be simplified as $\cdots$?


[1] Xiao, J. et al. Bridging the gap: Rademacher complexity in robust and standard generalization.
In 37th Annu. Conf. Learn. Theory, pp. 5074–5075. PMLR, 2024.

---

> ### Author Rebuttal · Authors · 2025-07-26
>
> We would like to thank the Reviewer for their kind words and helpful comments; we appreciate both.
>
> Regarding the Reviewer's questions:
>
> 1. *"The authors assume the FGSM attack, which is a weak attack in practice. It seems that the compared bound [1] did not assume the FGSM attack but used the precise maximal value."*
>
> Depending on the context, FGSM attacks can be less powerful than other known attacks, e.g., Carlini & Wagner. However, our choice of attack is motivated by two main reasons. Firstly, ADMM-DAD operates in a regression setting, where given input-output data, it aims to learn a nonlinear function solving the compressed sensing problem. Due to the nature of the problem at hand, we opt for an attack that does not explicitly rely on a classification paradigm (thereby rendering it quite irrelevant to our setting), but rather deploy an attack with broader utility; the characteristic of FGSM being based on the gradient of the loss function renders it as a good candidate for our study.
>
> Secondly, due to the appearance of the max operation in the definition of the adversarial generalization error, i.e., $$\tilde{\text{GE}}(h)=|\frac{1}{s}\sum\_{i=1}^s\max\_{\|\|\delta\|\_p\leq\varepsilon}\|\|h\_W(y\_i+\delta\_i)-x\_i\|\|_2^2-\mathbb{E}\_{(x,y)\sim\mathcal{D}}(\max\_{\|\|\delta\|\|\_p\leq\varepsilon}\|\|h\_W(y+\delta)-x\|\|\_2^2)|,$$ it is standard in related literature to work around this subtle point by relying on a surrogate loss function - in particular, of FGSM type. Similarly to [2], we rely on FGSM to discard the max operation and estimate a max-free adversarial generalization error: $$\tilde{\text{GE}}(h)=|\frac{1}{s}\sum\_{i=1}^s\|\|h_W(y_i+\delta_i^{\text{FGSM}})-x_i\|\|_2^2-\mathbb{E}\_{(x,y)\sim\mathcal{D}}\|\|h_W(y+\delta^{\text{FGSM}})-x\|\|\_2^2)|,$$ with $\delta^{\text{FGSM}}$ being the FGSM-based attack. Instead of working with a surrogate loss, the authors of [1] bypass the difficulty of the max operation by uniformly covering the perturbation set. Despite the fact of deploying standard covering numbers to estimate the adversarial generalization error of ADMM-DAD, we differentiate our approach from the one presented in [1] by turning to an FGSM surrogate loss, and then prove and use as a cornerstone of our study the Lipschitz continuity of the examined DUN w.r.t. the parameter matrix. This technical refinement leads to tighter upper bounds for the adversarial generalization error compared to the ones presented in [1].
>
> 2. "*It would be better to explain what the overcomplete sparsifier is."*
>
> A matrix $W\in\mathbb{R}^{N\times n}$ is a *sparsifier*/*sparsifying transform*, when its application on a given vector $x\in\mathbb{R}^n$ renders the latter sparse. The overcompleteness stems from the structure of $W$ per se, regarding the number of rows compared to the number of columns; that is, when $N>n$, we call $W$ overcomplete, following the intuition that its row vectors can form an "overcomplete" (not orthonormal) basis for the ambient space.
>
> 3. *"There are some typos: Line 120, $O_{n\times n}\in\mathbb{R}^n$
>  should be $O_{n\times n}\in\mathbb{R}^{n\times n}$. Line 169, it lacks a right bracket. Equation 10, why is there $log(exp(...))$, which can be simplified as $...$?"*.
>
> We thank the Reviewer for spotting the first two typos, we will correct them accordingly in the revised version of our manuscript. However, the expression $log(exp(...))$ is written as $log(exp\cdot(...))$ in the submitted manuscript, so that the parenthesis is multiplied by the exponential factor, and then the product is the input to the logarithm.
>
> [1] Xiao, J., et al., 2024. Bridging the gap: Rademacher complexity in robust and standard generalization. 37th Annu. Conf. Learn. Theory, pp. 5074–5075
>
> [2] Gao, Q., & Wang, X., 2021. Theoretical investigation of generalization bounds for adversarial learning of deep neural networks. J. Stat. Theory & Pract., 15(2), 51.

---

> > ### Comment · Reviewer_PrVf · 2025-08-05
> >
> > Thanks for your reply. It addresses my concerns. However, I decide to maintain my score due to the weak FGSM attack.

---

> > > ### Author Response · Authors · 2025-08-05
> > >
> > > We would like to thank the Reviewer once more for their kind words, positive feedback and time spent in the reviewing process, all of which we appreciate.
> > >
> > > We will incorporate all the Reviewer’s suggestions in the final version of our manuscript.

---

> ### Author Response · Authors · 2025-08-03
>
> We would like to thank the Reviewer again for their kind words, useful suggestions and effort put into the reviewing process. Given that the discussion phase is ending in a couple of days, we will be unable to participate in the discussion after that time. Hence, we wanted to reach out, to see if there are any remaining concerns after our rebuttal.
>
> We thank the Reviewer again for their input and time - we remain at their disposal if they require any further clarifications.

---

### Official Review · Reviewer_YdTx · 2025-06-25

**Clarity:** 2
**Significance:** 2
**Originality:** 3
**Rating:** 4
**Confidence:** 3

**Summary:**

This paper investigates the adversarial generalization of unfolding networks, where the adversarial perturbations are generated by the FGSM method, constrained in $l_2$-norm. They provide the adversarial generalization error bounds through the lens of adversarial Rademacher complexity and covering numbers. Experimental results on real-world data validate the theoretical findings.

**Questions:**

Refer to the weaknesses.

**Ethical Concerns:**

["NO or VERY MINOR ethics concerns only"]

**Final Justification:**

My main focus is on the improvement and scalability of the theoretical results, which have been adequately addressed in my opinion.

**Limitations:**

It is necessary to consider more general forms of $l_p$-norm adversarial attacks for arbitrary values of $p\geq 1$.

**Quality:**

2

**Strengths And Weaknesses:**

**Strength:**
+ This paper provides the first generalization analysis on the adversarial generalization of unfolding networks.
+ A series of experimental results corroborate the theoretical findings.

**Weakness:**
+ **Scalability of the theory.** This paper mainly focuses on $l_2$-norm constrained attacks. How difficult is it to extend the generalization analysis to the $l_p$ norm attack for arbitrary $p\geq 1$? Furthermore, FGSM is a well-known $\ell_\infty$-norm-based attack. It remains unclear why FGSM is specifically chosen to generate adversarial perturbations in this work, rather than employing other classical $l_2$-norm attacks such as the Carlini & Wagner (C&W) method. Please clarify this.
+ **The novelty of theoretical techniques.** What are the differences and improvements in the analytical techniques presented in this paper compared to those in (Xiao et al. 2021), especially for addressing outer maximization of adversarial losses?
+ **Comparison of the adversarial generalization bounds. (Table 1)** For a fair comparison, the proposed bounds should explicitly reflect the dependence on the norm of the parameter matrix. Additionally, the factor $\rm{Lip}_h^{L,\epsilon}$ should be given in the main text.
+ **Experiment.** There is a lack of a strong connection between the theoretical findings and the experimental section. In particular, the experimental section overlooks how the empirical error relates to the derived generalization bound. Is it possible to provide a direct comparison between the derived bounds and the observed generalization (e.g., augmenting some graphs with the values ​​suggested by the bounds)?
+ **Attack methods.** It would improve the experimental part if more attack methods were considered in this paper, such as PGD, C&W, etc.

The overall content of the paper is relatively dense, and the excessive use of symbols affects the overall readability.

Xiao et al. Bridging the gap: Rademacher complexity in robust and standard generalization. In 37th Annu. Conf. Learn. Theory, pp. 5074–5075. PMLR, 2024.

---

> ### Author Rebuttal · Authors · 2025-07-26
>
> We would like to thank the Reviewer for their valuable comments, which we appreciate.
>
> Regarding the Reviewer's questions:
>
> - *"**Scalability of the theory**. This paper mainly focuses on $l_2$-norm constrained attacks. How difficult is it to extend the generalization analysis to the $l_p$ norm attack for arbitrary $p\geq 1$? Furthermore, FGSM is a well-known $l_\infty$-norm-based attack. It remains unclear why FGSM is specifically chosen to generate adversarial perturbations in this work, rather than employing other classical $l_2$-norm attacks such as the Carlini & Wagner (C&W) method. Please clarify this."*
>
> We opt for an $l_2$ FGSM attack, as it is more relevant to the noise model of compressed sensing (CS). In the CS literature [1], with (Gaussian) noise added to the CS observations, i.e., $y=Ax+e$, it holds $\|\|e\|\|_2\leq\eta$, $\eta>0$; this detail is also found in Sec. A1 of our supplemental material. As we point out in lines 169-172 of our submitted manuscript, adversarially perturbing ADMM-DAD is equivalent to adding adversarial noise to $y$. Thus, it seems natural to transfer the noise model of standard CS to the adversarial setting. In fact, it is usual to investigate the robustness of models solving the CS problem against $l_2$, e.g., PGD, attacks [2-3]. However, $l_2$ attacks like C&W are crafted to promote a misclassification effect; this, in turn, is quite irrelevant to our setting, since ADMM-DAD aims to find a nonlinear function mapping $y$ to $x$, i.e., solves a regression task. Since FGSM is a state-of-the-art attack applicable in regression, and serves as a baseline for theoretical investigation and experimental comparison, we deem its choice, with under an $l_2$-norm, to be the most appropriate for our setup. Extending these results to different normed attacks would require a distinct treatment. For example, $p=1$ leads to a sparse attack, which requires to examine its support, in order to leverage the full strength of the adversary. Although this is a promising direction for future research, it lies beyond the scope of the present paper.
>
> - *"**The novelty of theoretical techniques**. What are the differences and improvements in the analytical techniques presented in this paper compared to those in (Xiao et al. 2021), especially for addressing outer maximization of adversarial losses?"*
>
> The outer-maximization shell of our approach is completely different to that of (Xiao et al. 2021), which considers the adversarial generalization error of a hypothesis class incorporating the max operation. To cope with the latter, the authors propose to uniformly cover the perturbation set appearing in that class. Although we also deploy covering numbers for estimating the adversarial generalization error, we follow an intrinsically different trajectory: we tackle the maximization by working with a surrogate loss [4-5]. Our key innovation compared to (Xiao et al. 2021) is that before employing covering numbers, we establish the Lipschitzness of the function implemented by the adversarially perturbed DUN, w.r.t. the parameter matrix. This Lipschitzness result, serving as a structural property, enables tighter adversarial generalization error bounds, via a refined covering number treatment and analysis. We describe this framework in lines 176-184 of the submitted manuscript, to better highlight our analytical technique.
>
> - *"**Comparison of the adversarial generalization bounds. (Table 1)** For a fair comparison, the proposed bounds should explicitly reflect the dependence on the norm of the parameter matrix. Additionally, the factor $\text{Lip}\_h^{L,\varepsilon}$ should be given in the main text."*
>
> We will revise our manuscript according to the Reviewer's suggestion, and incorporate the bound $\sqrt{\beta}$ of the spectral norm of the parameter matrix in Table 1, which will read as $\mathcal{O}(\sqrt{L\log(\beta\varepsilon)})$. Moreover, we will transfer the factor $\text{Lip}\_h^{L,\varepsilon}$, presently found as eq. ($\ddagger$) in Sec. C.2. of the supplemental material, to appear in Theorem 6 of Sec.3.
>
> - *"**Experiment**. There is a lack of a strong connection between the theoretical findings and the experimental section. In particular, the experimental section overlooks how the empirical error relates to the derived generalization bound. Is it possible to provide a direct comparison between the derived bounds and the observed generalization (e.g., augmenting some graphs with the values ​​suggested by the bounds)?"*
>
> From the theoretical perspective, we employ adversarial Rademacher complexity (ARC) - defined in eq. (9) of the submitted manuscript - of the hypothesis class, to provide a convincing partial answer to how adversarially generalizable DUNs are. ARC is a fundamental tool in studying adversarial generalization, via estimating the adversarial generalization error $\tilde{\text{GE}}(h)=|\mathcal{\tilde{L}}\_{\text{train}}(h)-\mathcal{\tilde{L}}\_{\text{true}}(h)|$, which in our FGSM context, is $$\tilde{\text{GE}}(h)=|(1/s)\sum_{i=1}^s\|\|h_W(y_i+\delta_i^{\text{FGSM}})-x_i\|\|_2^2-\mathbb{E}\_{(x,y)\sim\mathcal{D}}\|\|h_W(y+\delta^{\text{FGSM}})-x\|\|_2^2)|.$$ Adversarial generalization error bounds based on ARC hold uniformly over the hypothesis class, thereby establishing ARC as a worst-case complexity measure. We also apply techniques from asymptotic analysis to show that $\tilde{\text{{GE}}}(h)\leq\mathcal{O}(\sqrt{(NL\log(\varepsilon)/s)})$; this is presented in Corollary 4 of the submitted paper.
>
> From the practical perspective, a standard approach of empirically examining the generalization of neural nets in the standard learning regime [6] relies on approximating the true loss with the test loss. Similarly, we approximate $\mathbb{E}\_{(x,y)\sim\mathcal{D}}\|\|h_W(y+\delta^{\text{FGSM}})-x\|\|_2^2)$ with the adversarial test loss (given in eq. (19) of the submitted manuscript) $\tilde{\mathcal{L}}\_{\text{test}}(h)=(1/d)\sum\_{i=1}^d\|\|h(y'_i+\delta_i')-x'_i\|\|_2^2$, for test data $\{(y'_i,x'_i)\}\_{i=1}^d$. Then, the adversarial empirical generalization error $\tilde{\text{EGE}}(h)=\tilde{\mathcal{L}}\_{\text{test}}(h)-\mathcal{\tilde{L}}\_{\text{train}}(h)$, depicted in eq. (20) of the submitted manuscript and used to examine ADMM-DAD's adversarial generalization, approximates $\tilde{\text{{GE}}}(h)$. Given that we primarily aim to gain first theoretical and empirical insights in DUN adversarial generalization, our respective bounds do not need to be tight - although it is a promising future direction, mentioned in lines 327-329 of our submitted manuscript.
>
> Consequently, from the mathematical viewpoint, we employ the powerful notion of the ARC, albeit being a worst-case - complexity measure, to deliver adversarial generalization error bounds. From the practical side, we leverage the common - in standard learning literature - technique of approximating the true generalization error with its empirical counterpart, and adapt it accordingly for the adversarial learning regime, to experimentally elaborate on our theory. As such, there is no gap between our theoretical and our experimental results. Nevertheless, we understand that elaborating more on approximating the adversarial true error with the adversarial test error will better position our paper, so we will incorporate a version of the above discussion at the first revision opportunity, after eq. (20) found in Sec. 4.
> - *"**Attack methods**. It would improve the experimental part if more attack methods were considered in this paper, such as PGD, C&W, etc."*
>
> Given that the only well-known attacks suitable for regression are FGSM and PGD, and to be consistent with our theory and due to the costly nature of PGD, we preferred to adhere to FGSM for our experiments, illustrated in Figure 1 of the submitted paper. However, as we realise that perturbing ADMM-DAD with PGD would add value to our work, for the Reviewer's convenience, we indicate in the following table the results obtained after training and testing ADMM-DAD with $l_2$-norm PGD, e.g., with 10 iterations, for CIFAR10 (top) and SVHN (bottom):
> ```
>     |Clean MSE  | Adv MSE  |Adv. EGE×10⁻⁵
> ----|-----------|----------|---------
> ε\L      5/10      5/10       5/10
> .01 |.0287/.0185|.0294/.019 |.8/2
> .05 |.0302/.0234|.0311/.0242|1.9/1.5
> .1  |.0331/.0261|.034/.027 |2.6/1.8
> .5  |.035/.0264 |.0347/.028|4.2/4.5
> 1   |.036/.0281 |.037/.0292|5.7/6.3
> ----|-----------|----------|---------
>  ε\L  10/15       10/15      10/15
> .01 |.0135/.0102|.0148/.011|30/186
> .05 |.0149/.0113|.0151/.0129|38/203
> .1  |.0151/.0136|.0163/.0145|39/211
> .5  |.0142/.0128|.0145/.0137|86/218
> 1   |.016/.0145 |.0173/.0155|87/230
> ```
> Based on these values, we will create a plot similar to Figure 1, and add it in Sec. 4 of the revised manuscript.
>
> [1] Foucart, S., Rauhut, H., 2013. An invitation to compressive sensing. A mathematical introduction to compressive sensing, pp. 1-39
>
> [2] Raj, A., et al., 2020. Improving robustness of deep-learning-based image reconstruction. Int. Conf. Mach. Learn., pp. 7932-42
>
> [3] Wang, Y. et al., 2024. Benign overfitting in adversarial training of neural networks. Int. Conf. Mach. Learn.
>
> [4] Gao, Q., & Wang, X., 2021. Theoretical investigation of generalization bounds for adversarial learning of deep neural networks. J. Stat. Theory & Pract., 15(2), 51.
>
> [5] Yin, D., et al., 2019. Rademacher complexity for adversarially robust generalization. Int. Conf. Mach. Learn., pp. 7085-7094
>
> [6] Jakubovitz, D., et al., 2019. Generalization error in deep learning. Compressed sensing and its applications: 3rd Int. MATHEOn Conf., pp. 153-193

---

> > ### Comment · Reviewer_YdTx · 2025-08-04
> > **Official Comment by Reviewer YdTx**
> >
> > Thank you for your detailed and informative response. My main focus is on the improvement and scalability of the theoretical results, which have been adequately addressed in my opinion.

---

> > > ### Author Response · Authors · 2025-08-05
> > >
> > > We sincerely thank the Reviewer again for their encouraging remarks, constructive feedback, and the time dedicated to reviewing our work - all of which we greatly value.
> > >
> > > We will ensure that all of the Reviewer’s suggestions are included in the final version of our paper.

---

> ### Author Response · Authors · 2025-08-03
>
> We would like to thank the Reviewer again for their constructive feedback and time, both of which we appreciate. Since the discussion period is ending in a couple of days and we will be unable to participate in the discussion after that time, we wanted to reach out to see if there are any remaining concerns after our rebuttal.
>
> We thank the Reviewer again for their input and time - we remain at their disposal if they require any further clarifications.

---

### Official Review · Reviewer_mH3W · 2025-06-28

**Clarity:** 1
**Significance:** 2
**Originality:** 3
**Rating:** 4
**Confidence:** 3

**Summary:**

The paper presents a study that offers a theoretical analysis and understanding of unfolding networks under adversarial attacks. In particular, the authors address the problem of adversarial generalization by considering FGSM with an L2 norm. The main objective is to characterize the error bounds of the network in the presence of such attacks. The primary tool used for this analysis is the adversarial Rademacher complexity, which is employed to quantify model complexity under worst-case perturbations. Based on this, insights regarding the lipschitz properties and error bounds are derived. An experimental analysis is also provided to support the theoretical findings.

**Questions:**

Considering the weaknesses outlined above, the main questions I would like to raise for the authors are:

- Can you clearly address the main limitations of the proposed analysis from a practical perspective? Specifically, what are the concrete benefits and implications of this study in real-world settings? Additionally, could you clarify the practical significance of the assumptions made in your theoretical framework, and how restrictive they are in practice?

- Why is the analysis limited to FGSM with an L2 norm? FGSM is traditionally associated with the L-inf norm, as it relies on the sign of the gradient. Could you clarify what is meant by an “FGSM L2” attack in your setting? Did you intend to refer to a one-step L2 norm PGD attack without random initialization? Furthermore, could this analysis be extended to other adversarial settings (e.g. multistep attacks as PGD), which are also relevant and widely studied in the community?

**Ethical Concerns:**

["NO or VERY MINOR ethics concerns only"]

**Final Justification:**

I thank the authors for the very clear and constructive rebuttal. While my previous comments mainly concerned the presentation and the correctness and strength of the posed assumptions, the authors have clearly clarified these points in their response.
Considering these, my current score has improved to borderline accept.

**Limitations:**

As reported above, a more thorough discussion of the practical limitations of the work is required.

**Paper Formatting Concerns:**

No major formatting issues.

**Quality:**

3

**Strengths And Weaknesses:**

STRENGTHS

- The paper presents strong theoretical insights which, in my understanding, provide useful bounds that could help advance the understanding and potential certification of model robustness.
-Despite the complexity of the theoretical content, the soundness of the work appears to be respected.


WEAKNESSES

- The paper would benefit from a clearer introduction that better motivates the purpose of this theoretical study. While the analysis appears rigorous, a stronger contextual explanation of its practical relevance, particularly in the domain of DUNs, is needed. For example, in the first section, the authors should better clarify the importance of addressing from a practical perspective the question: "What is the generalization performance of DUNs in the presence of adversarial attacks?"

- After the bullet points listing the contributions in the introduction, the authors include a discussion that effectively resembles a "Related Work" section. However, it is currently embedded in the introduction, which makes the flow of the paper a bit more complex. In future versions, the authors should consider putting this part into a dedicated.


- Overall, the paper is difficult to follow. Although it is clear that the authors made a significant effort, a structure that could introduce the problem to a broader audience and highlights the practical implications of the findings would be appreciated.

- In Theorem 3, the authors introduce assumptions that should be further discussed from a practical perspective, specifically, how critical these assumptions are, and how they distinguish theoretical results from practical applicability. Additionally, it should be clarified why the proposed analysis is limited to FGSM with an L2 norm. Can it be extended to other adversarial settings?

---

> ### Author Rebuttal · Authors · 2025-07-25
>
> We would like to thank the Reviewer for their constructive comments, which we appreciate.
>
> Regarding the Reviewer's questions:
> - *"The paper would benefit from a clearer introduction that better motivates the purpose of this theoretical study. While the analysis appears rigorous, a stronger contextual explanation of its practical relevance, particularly in the domain of DUNs, is needed. For example, in the first section, the authors should better clarify the importance of addressing from a practical perspective the question: "What is the generalization performance of DUNs in the presence of adversarial attacks?"
> In Theorem 3, the authors introduce assumptions that should be further discussed from a practical perspective, specifically, how critical these assumptions are, and how they distinguish theoretical results from practical applicability. Additionally, it should be clarified why the proposed analysis is limited to FGSM with an L2 norm. Can it be extended to other adversarial settings?
> Can you clearly address the main limitations of the proposed analysis from a practical perspective? Specifically, what are the concrete benefits and implications of this study in real-world settings? Additionally, could you clarify the practical significance of the assumptions made in your theoretical framework, and how restrictive they are in practice?"*
>
> A core practical challenge of the proposed analysis is that it relies on Rademacher complexity, which is a worst-case complexity measure, leading to bounds holding uniformly over the hypothesis class. Still, despite the fact of being worst-case estimates, our adversarial generalization error bounds provide a convincing partial answer to how robust DUNs are in the presence of adversarial attacks. Importantly, they constitute significant theoretical baselines, as they can be leveraged to deliver tighter adversarial generalization error bounds - we leave this idea as a promising future work and briefly mention it in lines 327-329 of Sec. 5.
>
> Regarding the concrete benefits and implications of our study in real-world settings: CS, which DUNs are designed to address, finds application in domains, e.g., medical imaging, communications, cryptography, where safe and reliable inference is of utmost importance. For example, by leveraging the inherent overparameterization of DUNs to mitigate the effect of adversarial attacks, we provide stronger theoretical foundations for using DUNs in security-critical systems, protecting them against malicious manipulation. in fact, our result that overparameterization (via overcomplete sparsifiers) improves robustness offers a new training paradigm: instead of blindly increasing model size, one can strategically design DUNs to be efficient and resilient, thereby reducing reliance on trial-and-error adversarial training. All in all, to better position our paper w.r.t. the practical importance of addressing our main research question, we will add in the revised version the following paragraph, at the end of the contribution bullet points:
>
> *"Overall, our proposed study provides the first theoretical framework for understanding and improving the adversarial robustness of DUNs in CS, with direct implications for safety-critical applications. In medical imaging, for example, our results could help ensure that MRI reconstructions remain reliable even under adversarial perturbations, reducing the risk of misdiagnosis. Furthermore, by showcasing that overparameterization via learnable overcomplete sparsifiers improves robustness, our work offers concrete design principles for developing resilient, interpretable networks in real-world deployments, where accuracy, efficiency, and security must coexist."*
>
> On to the practicality of Assumptions (a) – (c): real-world images, e.g., CIFAR10 and SVHN, are naturally bounded in the $l_2$-norm. Then, we empirically observe the boundendess, w.r.t. the $l_2$-norm, by some constant $\mathrm{B}_\mathrm{out}>0$, of the output of the adversarially perturbed decoder. In fact, such boundedness assumptions for the input-output data are common when studying generalization [1-3]. Moreover, we are inspired by a recent study [4] stating that, from the applied perspective, even if the gradient of a ReLU-type activation - the soft-thresholding operator can be written as the sum of two shifted ReLUs - is zero, the training behaviour of the net is not affected. Finally, according to the code snippet creating an $l_2$ FGSM attack, e.g., from the state-of-the-art package cleverhans [5], developed for generating popular adversarial attacks, a division by zero is avoided. That is, for a model $f$ and input-output data $(Y,X)$, one can ensure that $\|\|\nabla_Y\ell(f(Y),X)\|\|_F\neq0$, with $\ell(\cdot)$ being the chosen loss; thus, the $l_2$ attack is well-defined.
> - *"After the bullet points listing the contributions in the introduction, the authors include a discussion that effectively resembles a "Related Work" section. However, it is currently embedded in the introduction, which makes the flow of the paper a bit more complex. In future versions, the authors should consider putting this part into a dedicated."*
>
> To increase readability of our manuscript, we will revise it to include all related work, starting from line 78 of the submitted manuscript, in a separate Section.
> - *"Overall, the paper is difficult to follow. Although it is clear that the authors made a significant effort, a structure that could introduce the problem to a broader audience and highlights the practical implications of the findings would be appreciated."*
>
> We appreciate the effort the Reviewer put on thoroughly reading our paper, so we address the aforesaid concern in detail in the answer of the 5th bullet-point.
> - *"Why is the analysis limited to FGSM with an L2 norm? FGSM is traditionally associated with the L-inf norm, as it relies on the sign of the gradient. Could you clarify what is meant by an “FGSM L2” attack in your setting? Did you intend to refer to a one-step L2 norm PGD attack without random initialization? Furthermore, could this analysis be extended to other adversarial settings (e.g. multistep attacks as PGD), which are also relevant and widely studied in the community?"*
>
> Our choice of an $l_2$ FGSM attack, instead of $l_\infty$, relies on two main reasons. Firstly, as is usually done in the compressed sensing (CS) literature [6], for the (Gaussian) noise $e$ added to the CS observations $y$, i.e., $y=Ax+e$, it holds $\|\|e\|\|_2≤\eta$, $\eta>0$ - this detail can also be found in Sec. A.1 of the paper's supplemental material. As we move from the model-based to the data-driven CS, one may notice that perturbing ADMM-DAD with adversarial noise tantamounts to adding adversarial noise to the CS observations. As such, it seems natural to constrain the adversarial noise in the $l_2$ ball, complying with the investigation of the adversarial robustness of models solving the CS problem [7-8]. What is more, we are inspired by relevant settings [9], where a model's adversarial generalization is addressed under $l_2$ FGSM attacks. Particularly, an $l_2$ FGSM attack $\delta$ pertains to normalizing the gradient of the loss to have $\varepsilon>0$ Euclidean norm. Overall, we stress our choice of an $l_2$ attack in lines 169-173 of the submitted manuscript. Due to the costly, iterative nature of PGD, and our aim to be consistent with out theoretical framework, we preferred to demonstrate only the results from FGSM, found in Figure 1 of the submitted manuscript. However, we trained and tested ADMM-DAD with an $l_2$ PGD attack, to showcase the adversarial robustness of ADMM-DAD against more powerful attacks. As we realise that also referring to the PGD, e.g., with 10 iterations, results would better position our paper, for the Reviewer's convenience, we present them in the table below, for CIFAR10 (top) and SVHN (bottom):
> ```
>     |Clean MSE  | Adv MSE  |Adv. EGE×10⁻⁵
> ----|-----------|----------|---------
> ε\L      5/10      5/10       5/10
> .01 |.0287/.0185|.0294/.019 |.8/2
> .05 |.0302/.0234|.0311/.0242|1.9/1.5
> .1  |.0331/.0261|.034/.027 |2.6/1.8
> .5  |.035/.0264 |.0347/.028|4.2/4.5
> 1   |.036/.0281 |.037/.0292|5.7/6.3
> ----|-----------|----------|---------
>  ε\L  10/15       10/15      10/15
> .01 |.0135/.0102|.0148/.011|30/186
> .05 |.0149/.0113|.0151/.0129|38/203
> .1  |.0151/.0136|.0163/.0145|39/211
> .5  |.0142/.0128|.0145/.0137|86/218
> 1   |.016/.0145 |.0173/.0155|87/230
> ```
> For illustrative purposes, we will create a plot - similar to Figure 1 - from the previous table, and place it in Sec. 4 of our revised manuscript.
>
> [1] Behboodi, A., et al., 2022. Compressive sensing and neural networks from a statistical learning perspective. Compressed Sensing in Inf. Process., pp. 247-277
>
> [2] Kouni, V., Panagakis, Y., 2025. Generalization analysis of an unfolding network for analysis-based Compressed Sensing. Appl. & Comput. Harmon. Anal., 101787
>
> [3] Joukovsky, B., et al., 2021. Generalization error bounds for deep unfolding RNNs. Uncertainty Artif. Intell., pp. 1515-1524
>
> [4] Bertoin, D., et al., 2021. Numerical influence of ReLU’(0) on backpropagation. Adv. Neural Inf. Process. Syst., 34, 468-479
>
> [5] Papernot, N., et al., 2016. Technical report on the cleverhans v2. 1.0 adversarial examples library. arXiv preprint arXiv:1610.00768
>
> [6] Foucart, S., Rauhut, H., 2013. An invitation to compressive sensing. A mathematical introduction to compressive sensing, pp. 1-39
>
> [7] Raj, A., et al., 2020. Improving robustness of deep-learning-based image reconstruction. Int. Conf. Mach. Learn., pp. 7932-42
>
> [8] Wang, Y. et al., 2024. Benign overfitting in adversarial training of neural networks. Int. Conf. Mach. Learn.
>
> [9] Farnia, F., et al., 2018. Generalizable adversarial training via spectral normalization. arXiv preprint arXiv:1811.07457.

---

> ### Author Response · Authors · 2025-08-03
>
> We would like to thank the Reviewer again for their constructive feedback and time, both of which we appreciate. Since the discussion period is ending in a couple of days and we will be unable to participate in the discussion after that time, we wanted to reach out to see if there are any remaining concerns after our rebuttal.
>
> We thank the Reviewer again for their input and time - we remain at their disposal if they require any further clarifications.

---

> > ### Comment · Reviewer_mH3W · 2025-08-04
> >
> > I thank the authors for the very clear and constructive rebuttal. While my previous comments mainly concerned the presentation and the correctness and strength of the posed assumptions, the authors have clearly clarified these points in their response. I would appreciate considering a switch towards a positive recommendation, with the hope that the authors will address the remaining presentation concerns and other minor issues in the final version.

---

> > > ### Author Response · Authors · 2025-08-04
> > >
> > > We would like to thank the Reviewer once more for their positive attitude, kind words, and useful comments that help to better position our submitted manuscript. We would be delighted to include all the Reviewer’s recommendations in the final version of our manuscript, as well as address the remaining presentation concerns and any other minor issues. Should the Reviewer believe there is anything else which could be addressed in the final version of our paper, we would be more than happy to add it as well.
> > >
> > > We thank the Reviewer again for their valuable time and the effort put in the reviewing process – we remain at their disposal if any further clarifications/additions are needed.

---

### Official Review · Reviewer_2Rop · 2025-07-02

**Clarity:** 4
**Significance:** 3
**Originality:** 4
**Rating:** 5
**Confidence:** 2

**Summary:**

This paper makes a theoretical contribution to the generalization of Deep Unfolding Network (DUN) under an adversarial setting. In particular, the paper focuses on a specific DUN, namely ADMM-DAD, solving the inverse problems in compressed sensing. The paper’s main result is bounding the adversarial generalization error bounds of ADMM-DAD. The paper presents several theoretical results, including the Lipschitz continuity of the ADMM-DAD decoder and an upper bound for the adversarial Rademacher complexity, to describe how the authors derived their main result. With experiments, the paper validates the theoretical findings.

**Questions:**

1. Please capitalize the title properly.
2. Figure 1 would look better if a log scale were used. In particular, the authors observed that the empirical generalization error (EGE) grows at the rate of $\sqrt{L\log(\varepsilon)}$, so the log-scale would show the relation between $\log(\varepsilon)$ and EGE better.
3. Is it possible to use more powerful attacks in the experiments? In particular, I’m curious about the robustness of ADMM-DAD against more powerful attacks. Because the attackers would consider attacks stronger than FGSM, such an investigation would add more practical value to this work.

**Ethical Concerns:**

["NO or VERY MINOR ethics concerns only"]

**Limitations:**

I don’t find any specific limitations or negative societal impacts of this work.

**Paper Formatting Concerns:**

This paper does not have any major formatting issues.

**Quality:**

3

**Strengths And Weaknesses:**

### Strengths
**Quality**: The experiments were performed on real-world image datasets, namely CIFAR-10 and SVHN, which is essential to demonstrate the practical value of the findings.

**Clarity**
1. The paper presents the theorems in separate boxes, which helps readers capture the main contributions.
2. Also, the paper emphasizes the meaning (or importance) of each theoretical finding. This also clarifies the key takeaways from each result.

**Significance**: The paper investigates the problem of adversarial learning in a relatively new application, i.e., DUN.

**Originality**: To the best of my knowledge, the paper presents novel ideas.

---
### Weaknesses
**Significance**
1. The contribution is limited to a particular application of DUN, i.e., the inverse problem of compressed sensing.
2. FGSM might be too weak to represent the solution that maximizes the error. However, I also agree that theoretical analysis becomes extremely difficult for more sophisticated attacks.

---

> ### Author Rebuttal · Authors · 2025-07-25
>
> We would like to thank the Reviewer for their kind words, we highly appreciate them.
>
> Regarding the Reviewer's questions:
> 1. *"The contribution is limited to a particular application of DUN, i.e., the inverse problem of compressed sensing."*
>
> Our choice of a compressed-sensing-tailored DUN is attributed to the fact that compressed sensing (CS) models a plethora of modern applications. For instance, CS is employed in secure data transmission and cryptographic protocols [1], communications in 5G networks [2], and tomographic imaging scans [3], to name but a few. In fact, all these applications must be addressed within a framework guaranteeing safe and reliable inference, to prevent catastrophic failures. Compared to other inverse problems, CS is particularly vulnerable to adversarial perturbations, due to its reliance on sparse and highly undersampled data, making it a natural stress test for robust model design. Notably, CS methodology has inspired the original paper introducing DUNs [4], with CS-based DUNs already being integrated into real-world CS pipelines, e.g., accelerated MRI in hospitals, where inference must be fast, interpretable, and trustworthy. Overall, due to its power of accommodating a wide range of applications, CS is an excellent paradigm showcasing the unfolding regime, with particular interest in how DUNs can be strategically designed to be both efficient and resilient.
>
> 2. *"Please capitalize the title properly."*
>
> We thank the Reviewer for the elegant suggestion; we will capitalize the title of our paper accordingly, to meet the conference's formatting.
>
> 3. *"Figure 1 would look better if a log scale were used. In particular, the authors observed that the empirical generalization error (EGE) grows at the rate of $\sqrt{L\log(\varepsilon)}$, so the log-scale would show the relation between $\log(\varepsilon)$ and EGE better."*
>
> We will encompass the Reviewer's suggestion in our revised manuscript, by using a log-scale in Figure 1, in order to better indicate the logarithmic scaling of our derived bounds.
>
> 4. *"FGSM might be too weak to represent the solution that maximizes the error. However, I also agree that theoretical analysis becomes extremely difficult for more sophisticated attacks. Is it possible to use more powerful attacks in the experiments? In particular, I’m curious about the robustness of ADMM-DAD against more powerful attacks. Because the attackers would consider attacks stronger than FGSM, such an investigation would add more practical value to this work."*
>
> Due to the fact that ADMM-DAD recasts the reconstruction problem of compressed sensing as a nonlinear regression one, we aim to deploy attacks of broader utility, rather than based on a classification paradigm. To that end, we choose FGSM, which relies on the gradient of the loss function and is a state-of-the-art attack, deployed for delivering adversarial generalization error bounds, as well as a baseline attack accommodating both classification and regression settings. Apart from FGSM, PGD-based attacks can also be used in a regression setting, so we harnessed time during the rebuttal period to conduct relevant experiments, in order to assess the adversarial robustness of ADMM-DAD against more powerful attacks - yet complying with regression. Due to our theoretical results being based on FGSM, we illustrated only these attacks; however, in order to highlight the potential of our model to be adversarially robust against different attacks, we provide in the table below the results from adversarially training and testing ADMM-DAD in the presence of an $l_2$-norm PGD, e.g., with 10 iterations, for CIFAR10 (top) and SVHN (bottom):
> ```
>      |Clean MSE  | Adv MSE  |EGE×10⁻⁵
> -----|-----------|----------|---------
> ε\L     |5/10      |5/10     |5/10
> .01 |.0287/.0185|.0294/.019 |.8/2
> .05 |.0302/.0234|.0311/.0242|1.9/1.5
> .1  |.0331/.0261|.034/.027 |2.6/1.8
> .5  |.035/.0264 |.0347/.028|4.2/4.5
> 1   |.036/.0281 |.037/.0292|5.7/6.3
> ----|-----------|----------|---------
>  ε\L |10/15     |10/15    |10/15
> .01 |.0135/.0102|.0148/.011|30/186
> .05 |.0149/.0113|.0151/.0129|38/203
> .1  |.0151/.0136|.0163/.0145|39/211
> .5  |.0142/.0128|.0145/.0137|86/218
> 1   |.016/.0145 |.0173/.0155|87/230
> ```
> In the first revision oppportunity, we will gather these results in a figure, similar to the results presented in Figure 1 of the submitted manuscript, to showcase the scaling of the test and empirical generalization errors, as the noise level of the PGD attack increases.
>
> [1] Yuge, N., et al., 2025. Privacy Preserving Deep Unrolling Methods and Its Application to Image Reconstruction. IEICE Trans. Fundam. of Electron., Commun. & Comput. Scie., 108(3), pp.304-312.
>
> [2] Xu, B., et al., 2024. Deep Unfolding Beamforming and Power Control Designs for Multi-Port Matching Networks. IEEE Trans. Wireless Commun.
>
> [3] Yang, L., et al., 2025. Distillation guided deep unfolding network with frequency hierarchical regularization for low-dose CT image denoising. Neurocomputing, p.130535.
>
> [4] Gregor K, LeCun Y. Learning fast approximations of sparse coding. Proc. 27th Int. Conf. Mach. Learn., pp. 399-406, 2010.

---

> ### Author Response · Authors · 2025-08-03
>
> We would like to thank the Reviewer again for their kind words, useful suggestions and effort put into the reviewing process. Given that the discussion phase is ending in a couple of days, we will be unable to participate in the discussion after that time. Hence, we wanted to reach out, to see if there are any remaining concerns after our rebuttal.
>
> We thank the Reviewer again for their input and time - we remain at their disposal if they require any further clarifications.

---

> > ### Comment · Reviewer_2Rop · 2025-08-05
> >
> > Thanks for the authors' response.
> >
> > I read the rebuttal, and the authors answered my questions properly in the rebuttal.
> > Particularly, I appreciate the additional experiment on PGD.
> >
> > However, I'll still maintain the same score, which is 5 (Accept)
> >
> > * I agree that compressed sensing has a wide range of applications; however, the scope is still limited to have a substantial impact.
> > * I disagree that FGSM is a state-of-the-art attack. It is probably true for attacking DUN, but most research in adversarial machine learning would not regard FGSM as a state-of-the-art attack.
> > * While the additional experiment showed that we may have more results on more powerful attacks (and PGD may represent a set of gradient-based attack algorithms well enough), this is not enough for the increased rating: 6, which is the highest rating.

---

> > > ### Author Response · Authors · 2025-08-06
> > >
> > > We thank very much the Reviewer once more for their positive evaluation and the thoughtful remarks.
> > >
> > > Indeed, we consider FGSM as a state-of-the-art attack in the context of the current paper.
> > >
> > > Thank you again for your time and constructive feedback, we appreciate both very much.

---

### Note · Authors · 2025-08-13

We sincerely thank the AC and Reviewers for their thorough reading, constructive feedback, and the positive assessments we received.

In view of the discussion so far, we trust and hope that our rebuttal and clarifications have addressed all the Reviewers' comments, and we look forward to incorporating all the relevant points of our discussion and exchanges at the first revision opportunity. These include replotting our figures in log-scale for enhanced visibility, providing more details on approximating the true error with the test error, fixing typos, and restructuring the introduction section to better highlight the practical importance of our framework and putting related work in a dedicated section for more clarity.

Finally, regarding the robustness of ADMM-DAD against PGD attacks, we will be glad to include the additional experiments that we shared in tabular form during the rebuttal period, and which answer the Reviewers' questions on this issue in the positive.

We hope that the discussion has reinforced the Reviewers' support of our paper, and we thank you again for your time and engagement.

---

### Decision · Program_Chairs · 2025-09-17

**Decision:**

Accept (poster)

**Comment:**

This paper presents the first theoretical analysis of adversarial generalization for unfolding networks, focusing on ADMM-DAD in compressed sensing and deriving adversarial error bounds via Rademacher complexity, supported by experimental validation. Reviewers found the contribution original and technically solid, with strengths in the clarity of theoretical results and experiments on real-world datasets. Main concerns prior to rebuttal included the reliance on FGSM (a relatively weak attack), limited generality beyond compressed sensing, and presentation issues such as a dense introduction and unclear motivation. The rebuttal clarified assumptions, justified the attack choice, and provided additional PGD experiments, which addressed most concerns. Overall, the work is recommended for poster acceptance, with suggested improvements in clarity, broader adversarial settings, and tighter theory–experiment integration.